# P38α MAPK-induced senescence in cranial suture progenitor cells promotes craniosynostosis
Zong Chen[1,2,3,8], Zhiyou Chen[4,8], Xinyan Chen[1], Yingying Yue[5], Yu Wang[1], Xueying Hou[6,7], Xiaoshuang Guo[1], Chenzhi Lai[1], Guodong Song[1] & Xiaolei Jin ®[1] ✉

Craniosynostosis is a congenital cranial developmental disorder that frequently leads to craniofacial deformities and even neurological dysfunction. The abnormalities in cranial suture progenitor cells (SPC) are considered a key event in craniosynostosis; however, the specific mechanism remains unclear. Using a syndromic craniosynostosis mouse model, we found that hyperactivation of p38α mitogen-activated protein kinase (MAPK) induced senescence in SPC of craniosynostosis mice. Integrated analysis of datasets from human patients and murine models, combined with cellular validation, revealed that p38/p53 activation and cellular senescence were prevalent across multiple forms of craniosynostosis and corresponding experimental models. Additionally, senescent cells significantly promoted osteogenic differentiation of SPC by paracrine Tgf-β1. Through in vivo and in vitro experiments, our evidence demonstrates that pharmacological inhibition of p38 MAPK, conditional knockout of *Mapk14*, and scAAV-mediated shRNA knockdown differentially attenuate SPC senescence, suture fusion, and elevated intracranial pressure, while ameliorating behavioral abnormalities in craniosynostosis mouse model. The present study supports p38α MAPK as potential therapeutic target for craniosynostosis.

Craniosynostosis refers to a condition characterized by craniofacial bone malformations caused by the premature fusion of one or more cranial sutures. The reported incidence rate is approximately 1 in 2500[1]. Owing to the premature fusion of cranial sutures, the extensibility of the membranous connections between the skull bones is disrupted, leading to compression of brain tissue by the affected skull. Severe cases may result in intracranial hypertension, hydrocephalus, delayed brain development, and intellectual disability[2]. In terms of appearance, the growth of the skull perpendicular to the fused cranial sutures is restricted, causing compensatory growth parallel to the closed sutures, which leads to cranial deformities.

Members of the fibroblast growth factor receptor (FGFR) family are the most commonly mutated genes associated with craniosynostosis[3]. These genes have been widely characterized and are fundamental to understanding the condition. *FGFR2* mutations lead to various syndromic craniosynostoses characterized by the fusion of coronal or multiple sutures,

including Crouzon, Apert, Pfeiffer, and Beare–Stevenson syndromes, with Crouzon syndrome being the most common[4,5]. A prominent feature of syndromic craniosynostosis is short-term fusion of cranial sutures (particularly the coronal sutures) and associated skull deformities[4]. Cranial sutures serve as niches of progenitor cells (SPC), which exhibit unique temporal and spatial distribution patterns and self-renewal capabilities[6]. The SPC plays a key role in skull injury repair and skull/suture development. Previous studies have established a close correlation between craniosynostosis and SPC abnormalities[7–10]. However, the detailed mechanisms remain poorly understood. Understanding the mechanisms driving SPC changes in the early stages of craniosynostosis is crucial for exploring skull and suture growth and development, and holds great potential for guiding the clinical treatment of craniosynostosis and skull injuries.

In this study, we provide evidence from a syndromic craniosynostosis mouse model showing that hyperactivation of p38α mitogen-activated

[1]Department of Craniomaxillofacial Surgery, Plastic Surgery Hospital, Chinese Academy of Medical Sciences & Peking Union Medical College, Beijing, China. [2]Department of Plastic Surgery, Peking Union Medical College Hospital, Chinese Academy of Medical Sciences & Peking Union Medical College, Beijing, China. [3]Center for Regenerative Medicine & Plastic Surgery Research, Peking Union Medical College Hospital, Beijing, China. [4]Sichuan Provincial People's Hospital, University of Electronic Science and Technology of China, Chengdu, China. [5]Jishuitan Hospital, Capital Medical University, Beijing, China. [6]Department of Neurosurgery, Shengjing Hospital of China Medical University, Shenyang, China. [7]Institute of Health Sciences, China Medical University, Shenyang, China. [8]These authors contributed equally: Zong Chen, Zhiyou Chen. ✉e-mail: jinxiaolei@psh.pumc.edu.cn

protein kinase (MAPK) induces SPC senescence, and Tgf-β1 secreted by senescent cells enhanced osteogenic differentiation of SPC, thereby promoting abnormal ossification and SPC exhaustion. Furthermore, our findings also demonstrated that p38/p53 activation and cellular senescence are prevalent in the cranial suture mesenchyme of both human craniosynostosis patients and murine models. Our evidence supports p38α MAPK as a potential therapeutic target for craniosynostosis, as pharmacological or genetic attenuation of p38α MAPK and conditional knockout (cKO) of *Mapk14* effectively prevents SPC senescence, mitigates suture fusion and craniofacial deformities in Crouzon mice. This study has demonstrated through comprehensive in vivo and in vitro experiments the critical role of p38α MAPK-mediated SPC senescence in regulating cranial suture homeostasis.

## Results

### Crouzon mice exhibit early postnatal coronal suture structural disorder, cell senescence, and depletion of mesenchymal components

In this study, stable *Fgfr2*[C361Y/+] Crouzon craniosynostosis mice were generated using CRISPR-Cas9 technology (Fig. S1a and S1b)[11]. Homozygous mutant mice did not survive embryonic development, whereas heterozygous mice exhibited symptoms similar to those of human Crouzon syndrome, characterized by facial shortening, brachycephaly, exophthalmos, and orbital hypertelorism (Fig. 1a). Fusion of the coronal sutures is a hallmark of Crouzon syndrome[12], and the mutant mice in this study consistently exhibited coronal craniosynostosis. μCT scanning and Hematoxylin-eosin (H&E) staining of cranial sutures revealed that, compared to wild-type (WT) mice, mutant-type (MUT) mice showed a sharp decrease in mesenchymal cells in the coronal sutures during the early postnatal period (7 days). By postnatal day 7 (P7), the mesenchymal components were nearly absent, cranial sutures appeared disordered, and trabecular connections had formed between the osteogenic fronts of the frontal and parietal bones. By postnatal day 14 (P14) to postnatal day 28 (P28), they were completely closed (Fig. 1b–d). By contrast, fusion of the sagittal sutures was relatively late. In MUT mice, the sagittal sutures showed trabecular connections on the dura mater side 1 month after birth and were completely closed 2–3 months after birth (Figs. 1e, S1c and S1d).

Clinical studies based on CT scans have found that coronal sutures naturally undergo physiological fusion during aging (>30–40 years)[13]. However, in mice, the cranial sutures remain unfused throughout life. As the mice aged (from 6 to 24 months), their coronal suture structures underwent notable changes, including a considerable decrease in mesenchymal cells (Fig. 1f). Based on these findings, we explored the relationship between suture fusion and cranial suture aging. Western blot analysis of senescence-related proteins (cell cycle inhibitory factors p53, p21, and p16) in the coronal sutures of WT and MUT mice showed that the expression levels of p21 and p53 protein in MUT mice were increased (Fig. 1g). Furthermore, we collected coronal suture tissues (coronal suture and 1 mm of tissue on each side) from WT and MUT mice for bulk RNA sequencing, and principal component analysis (PCA) revealed significant differences in gene expression between the two (Fig. S2a and S2b). Enrichment analysis indicated that p53, cellular senescence, and cell cycle–related pathways were enriched (Fig. S2c), whereas the expression of cell cycle inhibitory factors and cell differentiation-related genes was upregulated in MUT tissues (Fig. S2d). Gene set enrichment analysis (GSEA) further indicated that gene sets related to cell senescence and the p53 pathway were significantly upregulated in MUT mice (Fig. 1h and i).

Given that the activation of p53 is often closely related to cellular senescence and apoptosis[14], we investigated differences in the expression of senescence markers (p53, p21, and p16 protein) and the apoptosis index (cleaved caspase-3 protein) using immunostaining of coronal suture tissues of WT and MUT mice from P1 to P14, as well as WT mice aged 6–24 months. MUT mice showed a significant increase in p53 protein-positive cells (Fig. S3a and S3b) and p21-positive cells (Fig. 2a and c) in the coronal suture region. Notably, p21-positive cells were more widely

distributed near the proximal osteogenic fronts of the suture mesenchyme, whereas p16-positive cells were rare. However, no significant difference was noted in the apoptosis of coronal suture cells (cleaved caspase-3-positive cells) between the two groups (Fig. 2b and d).

Conversely, during mouse aging (from 6 to 24 months), the decrease in mesenchymal cell components within the coronal suture was mainly attributed to increased apoptosis rather than senescence (Fig. S3c–S3f).

Furthermore, we validated the senescence of cranial suture cells using Ki-67 immunostaining to assess cell proliferation and senescence-associated β-galactosidase (SA-β-Gal) staining in coronal suture tissues of WT and MUT mice (Fig. 2e and f). Early postnatal MUT mice exhibited decreased mesenchymal cell proliferation and increased SA-β-Gal-positive cells compared to those exhibited by WT mice (Fig. 2g and h). These findings suggest that proliferating cells in the coronal sutures of MUT mice exhibit reduced self-renewal capacity and other cellular manifestations of senescence.

### Prrx1 + SPC show increased senescence and osteogenic differentiation in the coronal sutures of MUT mice

SPCs are a heterogeneous group of progenitor cells (such as Gli1+, Axin2+, Ctsk+, and Prrx1+ cells) that establish the cranial sutures as their niche[15–18]. They have self-renewing and pluripotent characteristics and are crucial for the development and maintenance of cranial sutures. Based on the definition of SPC (CD51+ CD200+ CD105−) and their lineages described in previous studies[8,19], we used flow cytometry to assess the changes in the number of SPC in the coronal sutures of WT and MUT mice during the early postnatal period (P1–P10) (Figs. 3a and S4a). The proportion of SPC decreased with the growth of the mice, and the number of SPC of MUT mice showed an early progressive reduction (Fig. 3b and c). Next, we used fluorescence-activated cell sorting (FACS) to enrich the SPC population (CD51+ CD200+ CD105−) and verified their senescent phenotype at the cellular level. Cell cycle arrest is an important characteristic of senescence. First, we performed cell cycle analysis of SPC, and the results indicated that MUT SPC showed significant G1/G0 phase cell cycle arrest compared to that shown by WT SPC (Fig. 3d). Subsequently, we performed SA-β-Gal staining and comet assays to assess DNA damage in SPC. Senescence and DNA damage in SPC from the MUT group were significantly higher than those from the WT group (Fig. 3e–h).

Next, we clarified the differences in the osteogenic differentiation potential between the two groups of SPC. The osteogenic differentiation, measured by alkaline phosphatase (ALP) staining, showed that SPC of MUT mice exhibited higher levels of ALP than that of SPC from WT mice; however, no significant difference was noted in alizarin red staining (Fig. S4b). We evaluated the osteogenic differentiation of SPC from the coronal sutures using western blotting. GSEA indicated the upregulation of genes related to stem cell differentiation (Fig. S4c). Osteocalcin (Ocn) is a major marker of osteoblasts and can reflect the mineralization ability of bone tissue to a certain extent[20]. We performed immunostaining to assess osteogenic differentiation (ALP and Ocn) in the sutures of WT and MUT mice. The results supported the finding that osteogenic differentiation in the coronal sutures of MUT mice was enhanced in the early stage (P1–P7), and ALP+ cells were more distributed in the osteogenic fronts in MUT mice (Fig. 3i–k).

Gli1+, Axin2+, Ctsk+, and Prrx1+ cells were the main subgroups of SPC identified in previous studies[6,7,9,21]. To identify the subgroups of SPC that undergo senescence, we further performed immunostaining to mark Ctsk+ and Prrx1+ cell populations. The results indicated that part of Ctsk+ cells in MUT mice exhibited senescence (Fig. 3l), whereas senescent cells were mainly Prrx1+ cells (Fig. 3m). These findings suggest that senescence of SPC (Prrx1+ and/or Ctsk+ cells) in MUT mice may be the main cause of mesenchymal exhaustion in sutures.

### Activation of p53 mediated by hyperactivation of p38α MAPK induces phenotypic changes in SPC of MUT mice

Cellular senescence exhibits heterogeneity across different tissues or cells, and we confirmed that SPC senescence was manifested by p53/p21

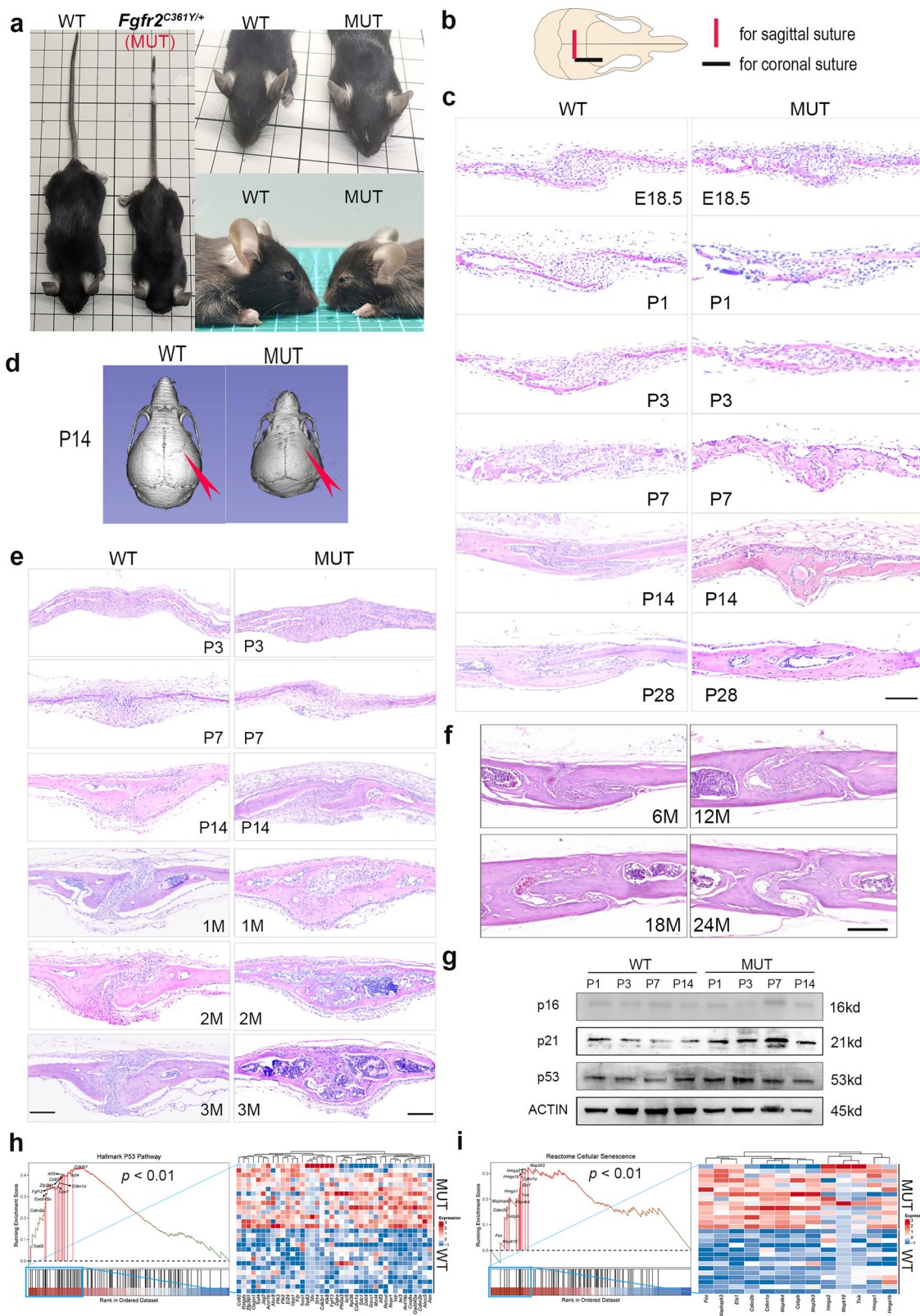

**Fig. 1 | *Fgfr2^{C361Y/+}* mice exhibit early depletion of coronal suture mesenchyme, fusion of cranial sutures, and senescence of mesenchymal cells. a** The overall appearance and frontal/lateral images of mutant-type (MUT) mice and wild-type (WT) mice. **b** The schematic diagram of histological sections of coronal and sagittal sutures. **c** The H&E staining of coronal sutures of MUT and WT mice (from embryonic day 18.5 (E18.5) to postnatal day 28 (P28)). Scale bar: 100 µm. $N$ = 6 sutures/group. **d** The µCT scanning images of skulls of MUT and WT mice at P14. $N$ = 5 mice/group. **e** The H&E staining of sagittal sutures of MUT and WT mice

(from postnatal day 3 (P3) to postnatal month 3 (3M)). Scale bar: 100 µm. $N$ = 4–8 sutures/group. **f** The H&E staining of coronal sutures of the 6-month-old (6M), 12M, 18M, and 24M WT mice. Scale bar: 200 µm. 6M, 12M, 24M, $N$ = 6 sutures/group; 18M, $N$ = 5 sutures/group. **g** Western blotting analyses of the coronal suture tissues of MUT and WT mice at 1, 3, 7, and 14 days after birth. **h**, **i** Gene sets enrichment analysis (GSEAs) specific to "Cellular Senescence" and "P53 Pathway" gene sets and heatmaps of core-enriched genes for bulk RNA sequencing data from coronal sutures of MUT mice and WT mice. $N$ = 14 per group.

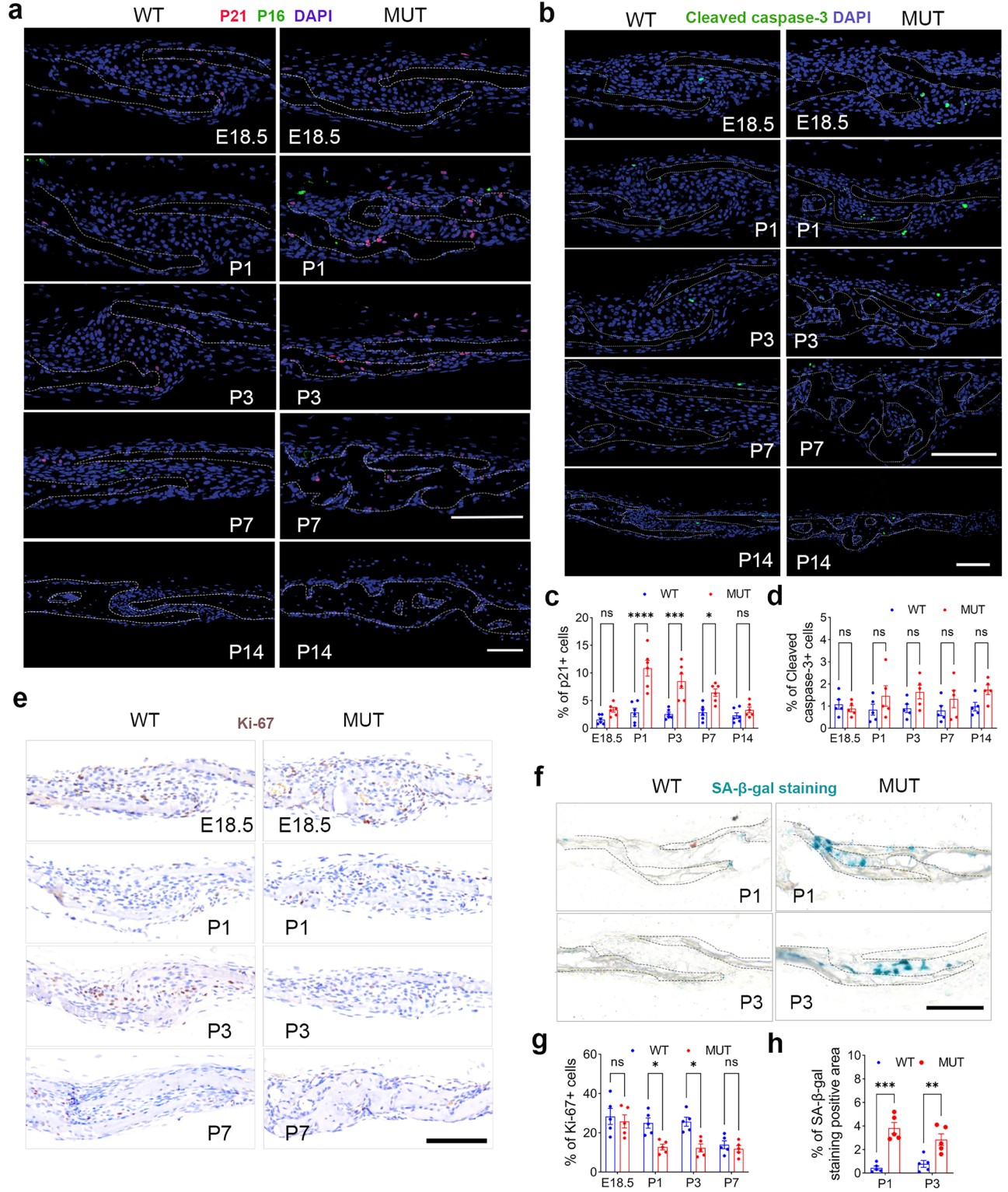

**Fig. 2 | Cellular senescence rather than apoptosis leads to depletion of mesenchymal components in coronal sutures of the MUT mice. a** The immunofluorescence images of p21 and p16 in coronal sutures of MUT and WT mice. p21 was in red, p16 was in green, and DAPI was in blue. Scale bar: 100 μm. **b** The immunofluorescence images of cleaved caspase-3 in coronal sutures of MUT and WT mice. Cleaved caspase-3 was in green, and DAPI was in blue. Scale bar: 100 μm. **c** and **d** Quantification of the proportion of p21-positive cells (**c**) and cleaved caspase-3-positive cells in coronal sutures of MUT and WT mice. *N* = 6 sutures/ group. **e** The immunohistochemical staining images of Ki-67 in the coronal sutures of MUT and WT mice. The cell nucleus was counterstained with hematoxylin. Scale bar: 100 μm. **f** The senescence-associated β-galactosidase (SA-β-Gal) staining images of MUT and WT mice coronal sutures. Scale bar: 100 μm. Quantification of the proportion of Ki-67 positive cells (**g**) and SA-β-Gal positive area (**h**) in coronal sutures of MUT and WT mice. *N* = 5 sutures/group. All data are presented as mean ± SEM. For **c**, **d**, **g**, and **f**, two-way ANOVA was used, followed by Sidak's test. *$P < 0.05$, **$P < 0.01$, ***$P < 0.001$, ****$P < 0.0001$. ns, not significant.

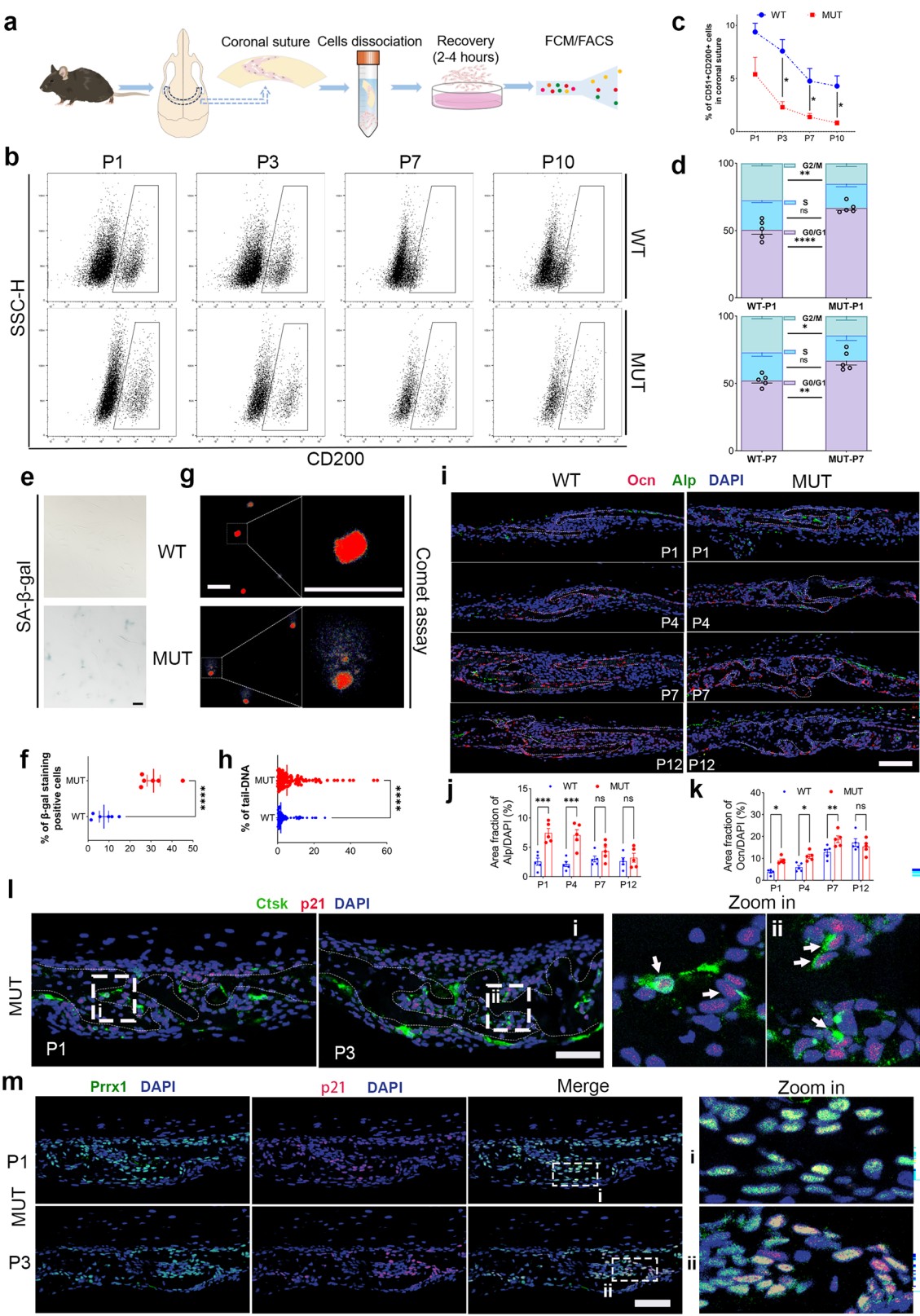

activation rather than p16. To further identify the key upstream targets that induce p53 activation and SPC senescence, we screened a siRNA sub-library related to p53 activation/regulation[22] and conducted a preliminary screening using the pp53-TA-luc (p53 activation reporter plasmid) and pRL-TK dual luciferase systems. Knocking down DNA damage-related genes and *Mapk14* (coding p38α MAPK protein) resulted in the greatest reduction in

p53 activation levels (Fig. 4a). Previous studies have confirmed the important role of Erk1/2 in mediating cranial suture ossification and craniosynostosis, and it is also closely related to cellular senescence. However, the present results of the siRNA sub-library screening experiment suggested that the attenuation of Erk1/2 (*Mapk3/Mapk1*) does not have a significant effect on the activation of p53 within SPC. Western blotting was used for

**Fig. 3 | The senescence of Prrx1+/Ctsk+ suture progenitor cells (SPC) is involved in the fusion of coronal sutures in MUT mice. a** Schematic diagram for identifying and isolating SPC through flow cytometry. The images of CD51+ CD200+ CD105− SPC from the MUT or WT mice (**b**), and the proportion of the SPC at P1 to P10 identified by flow cytometry (**c**). $N = 4$ per group. **d** Cell cycle analysis of the SPC derived from MUT and WT mice. $N = 5$ per group. The SA-β-Gal staining images of SPC (**e**) and the quantification of the number of staining positive cells (**f**) of MUT and WT mice. Scale bar: 50 μm. $N = 6$ per group. The representative images of comet assay of SPC (**g**) and the quantification of tail-DNA percentage of total DNA (**h**) of MUT and WT mice. Pi (in red) was used to stain the DNA. Scale bar: 50 μm. $N = 100$ nuclei per group. The immunofluorescence staining of Alp and Ocn (**i**), and the quantification of Alp (**j**) and Ocn (**k**) protein expression in coronal sutures of MUT and WT mice. Ocn was in red, Alp was in green and DAPI was in blue. Scale bar: 100 μm. $N = 5$ sutures/group. **l** The representative images of immunofluorescence staining with Ctsk (in green) and p21 (in red) in coronal sutures of MUT mice. The cell nucleus was counterstained with DAPI (in blue). Scale bar: 100 μm. $N = 5$ sutures/group. **m** The immunofluorescence images of prrx1 and p21 in coronal sutures of MUT and WT mice. Prrx1 was in green, p21 was in red, and DAPI was in blue. Scale bar: 100 μm. $N = 5$ sutures/group. Data are presented as mean ± SEM (data in (**h**) is presented as media). For **c**, **d**, **j**, and **k**, two-way ANOVA was used, followed by Sidak's test. For **f**, a two-tailed $t$-test was used. For **h**, the Mann–Whitney $U$-test was used. *$P < 0.05$, **$P < 0.01$, ***$P < 0.001$, ****$P < 0.0001$. ns, not significant.

secondary validation of the candidate targets (*Pdkdc, Atm, Atr, Mapk14*). With *Mapk14* knockdown, p53 levels in SPC from MUT mice significantly decreased (Fig. 4b). This suggests that the p38α MAPK/p53/p21 pathway may be involved in the senescence-associated phenotypic changes in SPC in MUT mice, which were validated at the protein level (Fig. 4c and d).

p38α MAPK plays an important role in bone formation and homeostasis, and its phosphorylation activation plays a key regulatory role in cell proliferation and differentiation[23]. Immunostaining of the suture suggested that the activation of p38α MAPK remained constant in the WT mice from embryonic stage to postnatal stage, which may be beneficial for maintaining bone and cranial suture homeostasis. However, in MUT mice, an increased activation of p38α MAPK was observed in the coronal sutures during the late embryonic stage (though not significant). After birth, a significant hyperactivation of p38α MAPK was observed, which was closely linked to p53 activation (Fig. 4e and f). GSEA for RNA sequencing further indicated significant upregulation of the p38 MAPK activation-related gene set in MUT mice (Fig. 4g). *Mapk11* (p38β), one of the important isoforms of *Mapk14* (p38α), is expressed in both bone tissue and mesenchyme[24]. To eliminate the contribution of p38β MAPK, we performed siRNA knockdown of *Mapk11* and *Mapk14* in MUT SPC. The results showed that *Mapk11* (p38β) did not interfere with the activation of the p53/p21 pathway (Fig. 4h).

### Inhibition of the p38α MAPK rescues and prevents MUT SPC from senescence, also impedes SPC osteogenic differentiation

Next, we investigated whether the inhibition of p38α MAPK and p53 could prevent or rescue the senescence of MUT SPC. We intervened with a p38α MAPK inhibitor (SB203580)[25] and/or a p53 inhibitor (pifithrin-α)[26]. We evaluated the intervention effect of p38α MAPK inhibitors on SPC (from P1 MUT mice) by observing the cell cycle distribution, expression levels of cell cycle and senescence-related proteins (pRb, p53, p21, etc.), and changes in cell proliferation levels (EdU assays) after inhibitor treatment. The results showed both inhibitors significantly mitigated G0/G1 phase cell cycle arrest, promoted Rb phosphorylation and normal cell cycle operation, increased SPC proliferation, and reduced the activation and accumulation of cell cycle inhibitory factors (Fig. 5a–e). Additionally, these inhibitors significantly reduced the number of SA-β-Gal-positive SPC (Fig. 5f) and impeded the osteogenic differentiation (by osteogenic differentiation induction and ALP staining) of MUT SPC (from P1 mice) (Fig. 5g). Next, we performed SA-β-Gal staining on SPC derived from P7 MUT mice, but different results were observed. p38α MAPK or p53 inhibitors did not significantly reduce the proportion of SA-β-Gal positive cells (senescent cells) (Fig. 5h). This suggests that the senescent phenotypes of SPC are reversible and salvageable only in the early stages. Previous evidence suggests that there is an interaction and regulatory relationship between Erk1/2 and p38 MAPK, and Erk1/2 has been confirmed to be significantly activated in Crouzon syndrome. In order to further investigate the effect of Erk1/2 on p38 MAPK/p53-mediated cellular senescence, we treated SPC with gradient concentrations of an Erk1/2-specific inhibitor (SCH772984) and then performed SA-β-Gal staining. The staining results indicated that SPC cell senescence did not depend on Erk1/2 activation (Fig. 5i).

### p38/p53 activation and cellular senescence are prevalent in the cranial suture mesenchyme of both human craniosynostosis patients and murine models

Given the established role of p38 in skeletal development and its mediation of SPC senescence in cranial suture homeostasis maintenance in Crouzon syndrome mice, we further analyzed RNA-seq data from craniosynostosis tissues and cells in public databases (human and murine) to investigate whether p38 MAPK-mediated cellular senescence is universally implicated in craniosynostosis pathogenesis. We performed GSEAs specific to gene sets of p38 MAPK activation, p53 activation and cellular senescence and quantification of core-enriched genes for the following RNA-seq data: (i) human non-syndromic craniosynostosis tissues (fused vs. patent sutures; GEO: GSE50796)[27]; (ii) human-induced pluripotent stem cells (iPSC)-derived mesenchymal stem cells (WT vs. *SNORD118*-deficient mutants; GEO: GSE223614)[28]; (iii) osteoblasts isolated from calvarial tissues of human isolated craniosynostosis patients and controls (GEO: GSE121780)[29]; (iv) coronal suture mesenchymal tissues from murine models (WT vs. *Fgfr2+/S252W* Apert syndrome mice, Facebase: FB00001172; WT vs. *Twist1+/−* Saethre-Chotzen syndrome mice, Facebase: FB00000902)[30]; and (v) facial skeletal tissues (mandibles) of WT and *Fgfr2+/S252W* Apert mice (GEO: GSE27976)[31].

The results of analysis revealed significant upregulation of gene sets associated with p38 MAPK activation, p53 activation, and cellular senescence across all craniosynostosis-derived human and murine samples compared to controls (Figs. 6a–d, S6a and S6b). Notably, the "Senmayo" gene set[32]—a previously validated signature for pan-tissue senescence identification, exhibited pronounced upregulation and enrichment in craniosynostosis samples, further corroborating the senescence phenotype.

Building upon these analytical findings, we further conducted cellular-level validation. Utilizing lentiviral vectors (*pLenti-Fgfr2_215_C361Y-EGFP* and control *pLenti-EGFP*, Fig. S6f) developed in our previous study[33] to mimic the *Fgfr2* mutation underlying Crouzon syndrome, we additionally engineered a *Twist1* knockdown lentiviral vector (*pLenti-Twist1_shRNA-EGFP*, Fig. S6c) to simulate the haploinsufficiency of *Twist1* mutation characteristic of Saethre-Chotzen syndrome[34]. Lentiviral plasmid designs and experimental workflows were detailed in Fig. 6e. Following lentiviral transduction of MC3T3-E1 cells (mouse embryo osteoblast precursor cells, which were established from the skull of C57BL/6 mice), stable transfectants were obtained through antibiotic selection for over 3 days and subsequently subjected to downstream analyses. Successful transfection was confirmed by EGFP fluorescence (Fig. S6d), and with *Twist1* mRNA levels quantified to verify knockdown efficiency (65%, Fig. S6e). Further analyses included western blotting to determine p-p38 protein levels across experimental groups, immunofluorescence staining to evaluate cellular p53 activation status, and SA-β-Gal staining to assess senescence. Results demonstrated that both *Fgfr2* mutations and *Twist1* knockdown (haploinsufficiency) significantly elevated p38 MAPK and p53 activation levels while promoting cellular senescence (Fig. 6f–h).

### The secretion of Tgf-β1 by senescent cells can significantly stimulate the osteogenic differentiation of adjacent SPC

The SASP is an important component of cellular senescence. Previous studies have shown that senescent cells can induce senescence and functional

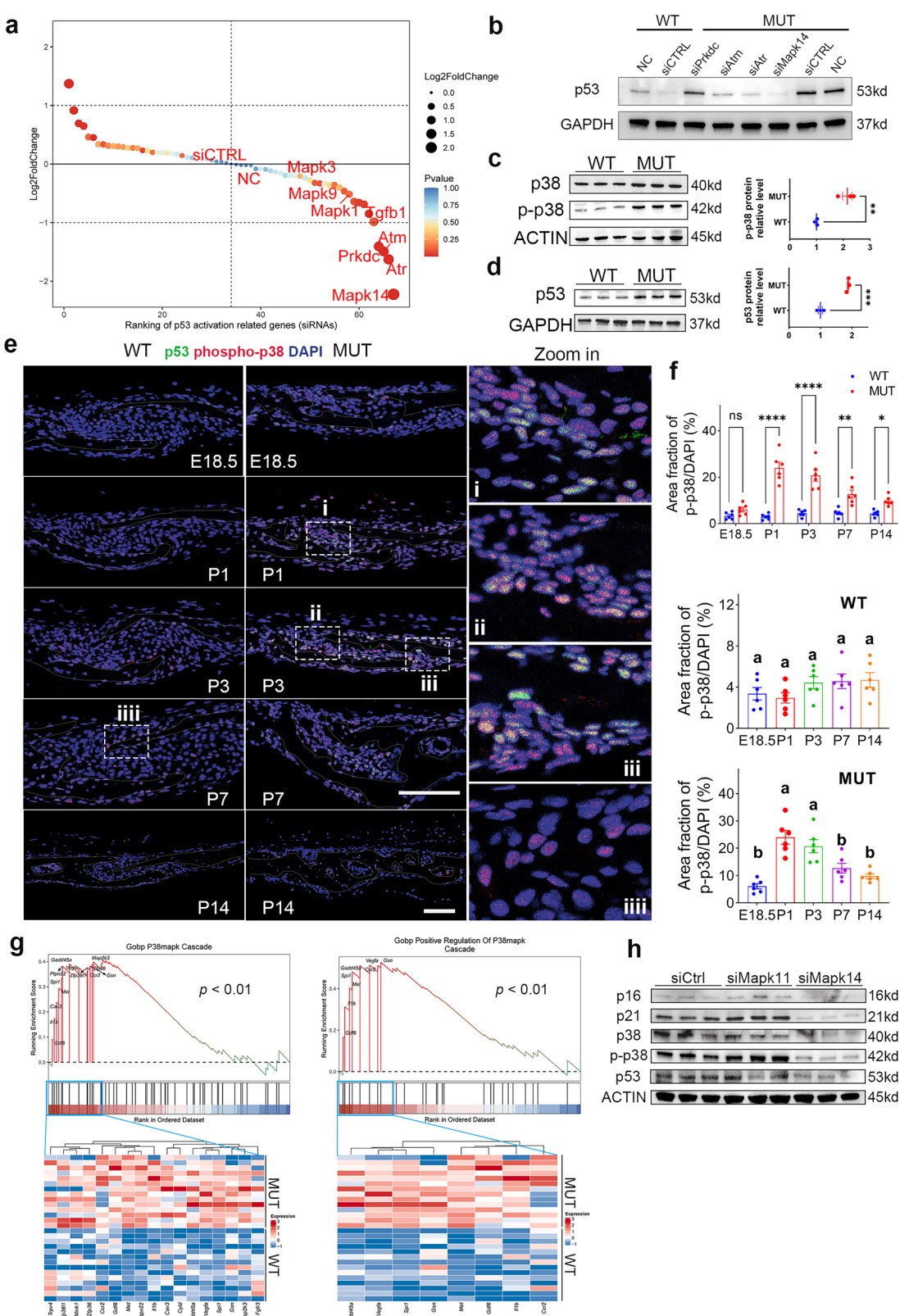

abnormalities in surrounding cells by paracrine SASPs[35–38]. Based on RNA sequencing results, we screened for relevant SASPs (Fig. S6a). We then validated several classical SASPs based on previous studies[37]. The results showed that the mRNA levels of SASP-associated genes such as *Tgfb1*, *Vegfa*, and *Cxcl12* were significantly upregulated in MUT SPC (Fig. S6b), and inhibition of p38α MAPK could reduce the transcription level of these SASP mRNA in MUT SPC (Fig. S6c). Furthermore, we quantified the protein secretion level of key SASPs in the supernatant of cell culture. The results indicated that the level of Tgf-β1 protein secreted by MUT SPC was significantly increased (5-fold) compared to WT SPC, and could be suppressed by p38α MAPK/p53 inhibitors (Fig. 7a). We also observed a significant increase in Tgf-β1 secretion in the cranial sutures of MUT mice (Fig. 7b).

**Fig. 4 | Hyperactivation of p38α MAPK drives p53/p21-dependent senescence of MUT SPC. a** Ranking scatter plot of siRNA sub-library screening related to p53 activation of MUT SPC. siCTRL, treated with control siRNA. NC, with no treatment. *N* = 5 plates/group. **b** Western blotting analyses of candidate targets for second round validation. siCTRL, treated with control siRNA. NC with no treatment. **c** Western blotting analyses of p38 and phospho-p38 (p-p38) protein and the quantification of protein expression of MUT and WT SPC. *N* = 3. **d** Western blotting analyses of p53 protein and quantification of the protein expression of MUT and WT SPC. *N* = 3. The immunofluorescence images of p-p38 and p53 (**e**), and the quantification of p-p38 (**f**) protein expression in coronal sutures of MUT and WT mice. p-p38 was in red, p53 was in green, and DAPI was in blue. Scale bar: 100 μm. *N* = 6 sutures/group. **g** GSEAs specific to "P38MAPK Cascade" and "Positive Regulation of P38MAPK Cascade" gene sets and a heatmap of core-enriched genes for bulk RNA sequencing data from coronal sutures of MUT mice and WT mice. *N* = 14 mice/group. **h** Western blotting analyses of p38/p53/p21 related protein expression of MUT SPC with siRNA treatment. *N* = 3. All data are presented as mean ± SEM. For **c** and **d**, two-tailed *t*-test was used. For **f**, two-way ANOVA and one-way ANOVA were used, followed by Sidak's test and Tukey's test, respectively. *$P < 0.05$, **$P < 0.01$, ***$P < 0.001$, ****$P < 0.0001$. ns, not significant. Alphabet method was also used to indicate significance.

To clarify the impact of key SASPs on adjacent (non-senescent) SPC, we simulated the in vivo paracrine mode and constructed a co-culture model of WT SPC and MUT SPC (Fig. 7c). After co-culturing for 3, 5, and 7 days, we collected the lower chamber SPC for SA-β-Gal and ALP staining to investigate whether senescent SPC (MUT SPC) could induce WT SPC senescence or affect osteogenic differentiation through SASP. The experimental results showed that SASP derived from senescent SPC could significantly increase the osteogenic differentiation level of WT SPC (co-cultured for 5 or 7 days), but did not induce senescent changes in WT SPC (until co-cultured for 7 days) (Fig. 7d–g).

Based on the above results, we speculated that senescent cells induced and enhanced osteogenic differentiation of adjacent SPC by secreting Tgf-β1. We treated WT SPC with a gradient concentration of Tgf-β1 protein and performed ALP staining. We found that treating SPC with 1 and 5 ng/ml Tgf-β1 and culturing for 5 days significantly enhanced osteogenic differentiation levels (Fig. 7h–j). Subsequently, we conducted rescue experiments based on the co-culture model, and the results showed that both p38α MAPK inhibition (SB203580) and Tgf-β1 inhibition (SB525334, a specific Tgf-β1 Receptor inhibitor) could significantly rescue the effect of SASP secreted by senescent cells on the osteogenic differentiation of SPC (Fig. 7k and l). To sum up, our findings suggest that Tgf-β1 may be a key mediator (paracrine effect) for abnormal ossification and fusion of the cranial sutures in MUT mice.

## Pharmacological, genetic attenuation or conditional knockout of p38α MAPK mitigates craniosynostosis and cranial deformities in Crouzon mice

The in vitro calvarium explant culture model, established in previous studies[39], provides a robust experimental platform for exploring the effect of drug interventions on craniosynostosis (Fig. 8a). Our findings support the key role of SPC senescence and SASP in the fusion of sutures in Crouzon mice. Using the explant culture model, we investigated the effects of drugs (SB203580, pifithrin-α, and SB525334) targeting p38α MAPK/p53-mediated SPC senescence on the fusion of sutures in MUT mice. The drug interventions mitigated the fusion of coronal sutures to varying degrees (Fig. 8b) and restored the number of SPC (Fig. 8c). We further tested the effects of different concentrations of Tgf-β1 protein (0, 0.1, 1, and 10 ng/ml) on calvarial explants from wild-type mice, and the results suggested that simply increasing local Tgf-β1 concentration was insufficient to induce structural fusion of the sutures (Fig. S7a).

Subsequently, we conducted animal experiments by administering different doses of the drugs (SB203580, pifithrin-α, or SB525334) via daily scalp injections in postnatal mice until P14. Tissue staining was performed on coronal sutures at P14 and P28 to evaluate the effects of the interventions (Fig. 8d). The p38α MAPK and Tgf-β1 specific inhibitors mitigated the fusion of coronal sutures to varying degrees (Fig. 8e). In addition, drug effectiveness increased with higher dosages; however, high doses resulted in the mortality of young mice, limiting the feasibility of further dosage increases.

Due to limitations such as instability, tissue irritation, and damage caused by local drug injection, as well as off-target effects of drugs, the effectiveness evaluation of targeted therapy was weakened. Next, we investigated genetic intervention targeting p38α MAPK (*Mapk14*). Given that *Mapk14* is essential for the growth and development of mice,

homozygous *Mapk14* KO mice die during the embryonic stage, consistent with previous studies[23]. We generated *Mapk14*^−/+ mice and *Fgfr2*^C361Y/+; *Mapk14*^−/+ mice (Fig. S7b). μCT scanning and histological analysis revealed that the heterozygous knockout of Mapk14 did not significantly mitigate coronal suture fusion or skull deformities in MUT mice (Fig. S7c); however, it mitigated bite deformities (Fig. S7d).

To minimize off-target effects while avoiding lethality, we generated *Fgfr2*^C361Y/+; *Prrx1*^cre+; *Mapk14*^f/+ (*Mapk14*^f/+ cKO) and *Fgfr2*^C361Y/+; *Prrx1*^cre+; *Mapk14*^f/f (*Mapk14*^f/f cKO) mice to conditionally knockdown or knockout *Mapk14* (p38α MAPK) in Prrx1+ cells, with breeding and experimental protocols detailed in Fig. 8f. Histological analysis of coronal sutures collected from WT and cKO mice at P1, P7, P14, and P28 revealed that *Mapk14*^f/+ cKO mice showed no significant alleviation in premature suture fusion, whereas *Mapk14*^f/f cKO mice maintained patent sutures (Fig. 8g), indicating that *Mapk14* ablation in Prrx1+ cells effectively ameliorated craniosynostosis, with concomitant resolution of cranial deformities as confirmed by μCT scanning (Fig. 8h). Subsequent SA-β-Gal staining and osteogenic differentiation assays (Alizarin Red staining) revealed that *Mapk14* deletion significantly rescued SPC senescence while partially suppressing osteogenic potential (Fig. 8i and j), consistent with μCT findings showing inhibited physiological fusion of the postfrontal suture in Mapk14f/f cKO mice. Western blotting of SPC isolated from sutures of *Mapk14*^f/f cKO mice demonstrated a fourfold reduction (by gray value of strips) in p38 protein levels compared to WT counterparts (Fig. 8k).

Previous studies have highlighted the role of p38 MAPK in bone development[23], and our findings suggest that a specific level of p38α MAPK activity is crucial for early suture development. Self-complementary adeno-associated viruses (scAAVs) loaded with short hairpin RNAs (shRNAs) are a type of stable, rapid, and efficient genetic attenuation therapy[40]. We then used scAAV9-shRNA_*Mapk14*-EGFP (scAAV9-shRNA_Scramble-EGFP as a control) injected subcutaneously into the scalp of postnatal mice to efficiently attenuate p38α MAPK hyperactivation in Crouzon mice (Figs. S8a, b and 9a, b). Silencing *Mapk14* significantly improved skull deformities in Crouzon mice (Fig. 9c). Furthermore, μCT scanning and histological analysis confirmed that targeting *Mapk14* significantly mitigated craniofacial deformities in Crouzon mice, preventing suture fusion (Fig. 9d). However, targeting *Mapk14* in WT mice impaired coronal suture development to some extent, resulting in reduced osteogenesis in the coronal sutures (Fig. 9d). Elevated intracranial pressure (ICP) is one of the most common complications of craniosynostosis. Long-term elevated ICP can cause serious consequences such as hydrocephalus, abnormal brain development, and even cerebral herniation[41]. Our results showed that scAAV-sh*Mapk14* attenuation therapy significantly reduced elevated ICP in MUT mice (mainly by preventing cranial suture fusion) (Fig. 9e and f). Our preliminary research has shown that Crouzon mice exhibit typical behavioral abnormalities. Through the rotarod test and novel object test, the results showed that attenuation therapy also significantly improved the behavioral abnormalities of MUT mice (Fig. 9g–j).

## Discussion

Approximately 20% of all known craniosynostotic diseases are caused by mutations in the *FGFR* family members[3]. Among these, *FGFR2* mutations are associated with the onset of various types of craniosynostosis, with Crouzon syndrome being the most common, and bilateral coronal

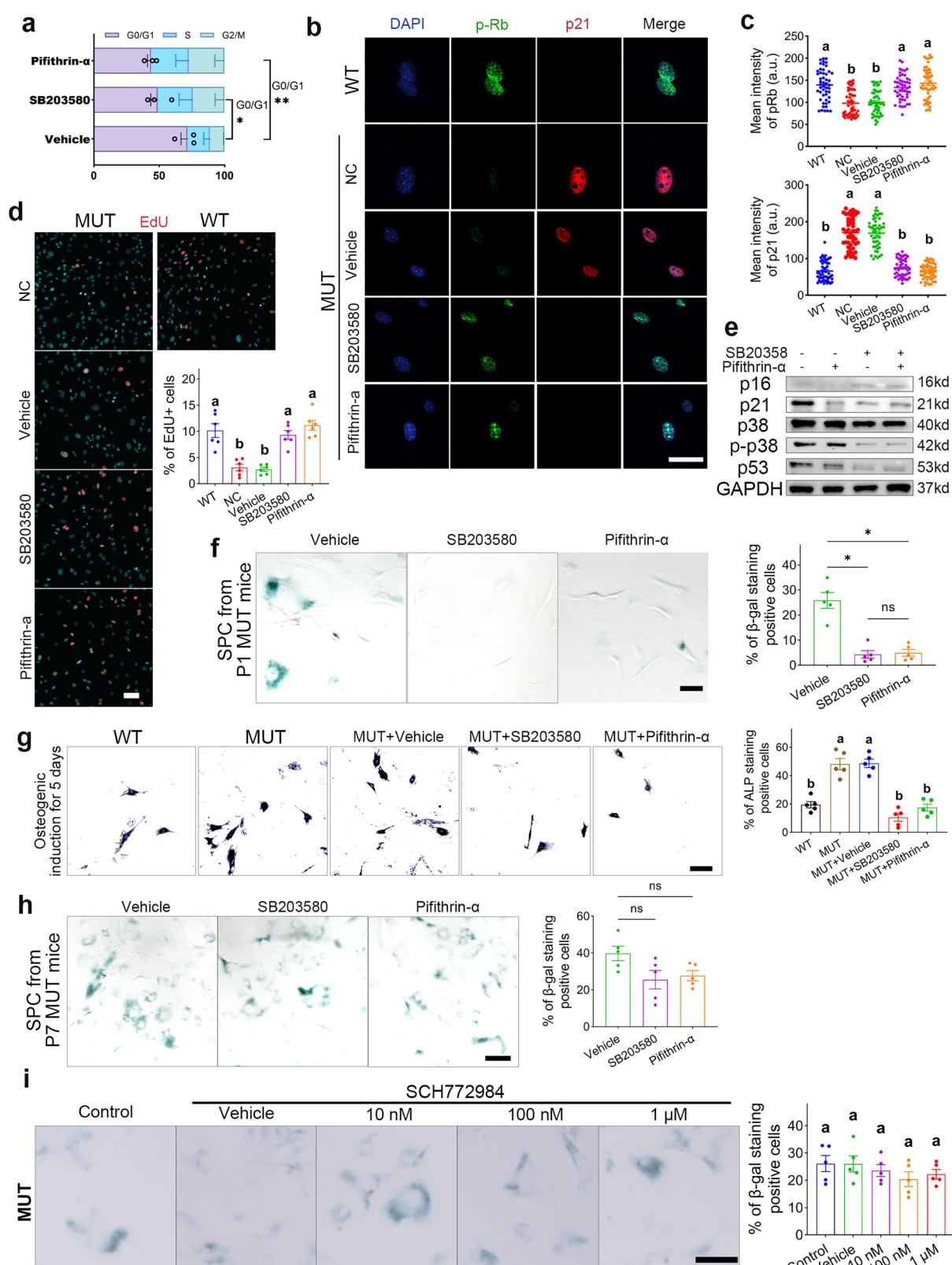

craniosynostosis serving as one of the main features of syndromic craniosynostosis[4]. The abnormalities of function/proportion of progenitor cells (SPC) within cranial sutures were considered as the key event in craniosynostosis[6]. Understanding the mechanisms underlying SPC changes in craniosynostosis is crucial for advancing treatments and research related to craniosynostosis and skull injury repair. p38 MAPK can activate

numerous transcription factors, including p53, through phosphorylation and plays important regulatory roles in cell proliferation, differentiation, apoptosis, and senescence[42]. Previous studies have suggested an increase in p38 MAPK activation in the skull of mice with craniosynostosis, but the effects and mechanisms of this on SPC and the progression of craniosynostosis are unclear[43,44]. In the present study, using a Crouzon mouse model,

**Fig. 5 | Inhibition of p38α MAPK effectively mitigates and rescues SPC senescence, but inhibits the osteogenic differentiation of SPC. a** Cell cycle analysis of MUT SPC treated with vehicle, p38 MAPK inhibitor, and p53 inhibitor. $N = 3$. **b** The representative images of immunocyte fluorescent staining with phospho-Rb (p-Rb, in green) and p21 (in red) of SPC with/without inhibitor treatment. Nuclei were stained with DAPI (in blue). NC, with no treatment. Scale bar: 50 μm. **c** Quantification of the p-Rb and p21 protein expression of SPC with/without inhibitor treatment. $N = 50$ cells/group. **d** EdU staining of SPC and the quantification of EdU (in red) positive cells number with/without inhibitor treatment. Nuclei were stained with DAPI (in blue). Scale bar: 50 μm. $N = 6$ per group. **e** Western blotting analyses of p38/p53/p21 related protein expression of MUT SPC with/without inhibitor treatment. **f** The SA-β-Gal staining images of MUT SPC (from P1 mice) with/without inhibitor treatment and the quantification of staining positive

cells number. Scale bar: 50 μm. $N = 5$ per group. **g** The ALP staining images of SPC with/without inhibitor treatment and osteogenic induction, and the quantification of staining-positive cells number. Scale bar: 50 μm. $N = 5$ per group. **h** The SA-β-Gal staining images of MUT SPC (from P7 mice) with/without inhibitors treatment and the quantification of staining positive cells number. Scale bar: 100 μm. $N = 5$ per group. **i** The SA-β-Gal staining images of MUT SPC with/without gradient concentrations of an Erk1/2-specific inhibitor (SCH772984) and the quantification of staining positive cells number. Scale bar: 100 μm. $N = 5$ per group. All data are presented as mean ± SEM. For **a**, one-way ANOVA was used, followed by Dunnett's test. For **c**, **d**, **f**, **g**, **h**, and **i**, one-way ANOVA was used, followed by Tukey's test. $*P < 0.05$, $**P < 0.01$, $***P < 0.001$, $****P < 0.0001$. ns, not significant. Alphabet method was also used to indicate significance.

we demonstrated through both in vitro and in vivo experiments that p38α MAPK hyperactivation promotes suture fusion by inducing SPC senescence. Integrated analysis of RNA-seq data from multiple human craniosynostosis patients and murine models demonstrates that p38 activation, p53 activation, and cellular senescence are consistently upregulated and pathologically prevalent in craniosynostosis-derived tissues/cells compared to WT controls. Subsequent experiments showed that senescent cells significantly promote osteogenic differentiation of SPC by paracrine Tgf-β1. Furthermore, pharmacological, genetic attenuation, or cKO of p38α MAPK (*Mapk14*) effectively prevents SPC senescence and suture fusion, alleviating cranial deformities in Crouzon mice.

Previous imaging studies[45–47] have consistently demonstrated that during human aging (beyond 40-60 years of age), physiological mesenchymal depletion and fusion occur in multiple cranial sutures (e.g., coronal, sagittal, and lambdoid sutures). Our histological evidence from murine aging models revealed analogous physiological mesenchymal depletion in cranial sutures (albeit without suture fusion), suggesting that pathological suture fusion may represent a localized senescent pathology. Integrated analysis of RNA-seq data from human craniosynostosis-derived suture tissues/mesenchymal cells and murine coronal suture mesenchyme demonstrated that cellular senescence is universally implicated in the pathogenesis of diverse craniosynostosis subtypes across species (we further validated it through lentivirus transfection experiments with MC3T3-E1 cells).

Gli1+, Axin2+, Ctsk+, and Prrx1+ cells, which mark the SPC and were identified in previous studies, with Prrx1+ SPC being the predominant group[15,21]. Previous work by Liu et al.[16] demonstrated through genetic lineage tracing and cell depletion assays that Prrx1-expressing cells in mice function as stem cells for bone, white adipose tissue, and dermis, being indispensable for tissue homeostasis and repair. Aldawood et al.[21] quantitatively demonstrated through single-cell RNA-seq analysis that in postnatal murine cranial sutures, a substantial proportion of cells (>40%) expressed Prrx1 and Ctsk. Their study provided the first groundbreaking demonstration of significant overlap between Prrx1+ cells and Ctsk/Gli1/Axin2-expressing populations. Strikingly, in situ hybridization of 80-day post-conception human fetuses further revealed broad PRRX1+ cell distribution in developing human sutures, while clinical genomic analyses established a strong pathogenic correlation between PRRX1 haploinsufficiency and human craniofacial malformations. Our research indicated that Prrx1+ SPC, along with a certain amount of Ctsk+ SPC, exhibited clear signs of cellular senescence in the sutures of MUT mice. Given the predominance of Prrx1+ cells within the cranial suture mesenchymal cell population and their significant overlap with other suture stem cell populations, this study employed Prrx1+ cell-specific *Mapk14* cKO to prevent SPC senescence and ameliorate craniosynostosis. Recent seminal work by Greenblatt et al.[7] demonstrates that cranial suture homeostasis and pathological fusion are governed by the balance between two distinct stem cell populations—Ctsk+ stem cells and Ddr2+ stem cells. Their findings indicate that pathological depletion of Ctsk+ stem cells derepresses Ddr2+ stem cells, leading to aberrant expansion of the latter and driving pathological ossification. Given the substantial overlap between Prrx1+ and Ctsk+ cells[21], one speculation is that cellular senescence constitutes a critical

intermediate link connecting Ctsk+ stem cell depletion with aberrant Ddr2+ stem cell activation. For instance, SASP released by senescent Ctsk+ stem cells or their neighboring cells may directly establish a pro-inflammatory and pro-fibrotic microenvironment that pathologically activates the osteogenic potential of Ddr2+ stem cells. In other words, p38 MAPK–p53 pathway-mediated cellular senescence and Ddr2+ stem cell-mediated endochondral ossification may not represent independent pathogenic pathways but rather function as synergistic and mutually amplifying coupled mechanisms during craniosynostosis pathogenesis.

In this study, we identified a key SASP—Tgf-β1, in senescent SPC. The secretion of Tgf-β1 was dependent on senescent cells and p38α MAPK. By constructing a co-culture model of WT SPC and MUT SPC, our evidence suggested that senescent SPC can significantly enhance the osteogenic differentiation of adjacent SPC through paracrine Tgf-β1. Accumulating evidence positions Tgf-β1 as a universal molecular bridge connecting cellular senescence to disrupted tissue homeostasis[48,49]. In the pulmonary system, Tgf-β1 signaling has been established as a central mechanism driving alveolar type II (AT2) epithelial cell senescence and pulmonary fibrosis[50]. Within the cardiac context, Tgf-β1 signaling promotes cardiomyocyte senescence through modulation of specific epigenetic modifications, particularly H4K20me3 status[51]. Furthermore, in both skeletal muscle and nervous systems, elevated Tgf-β1 signaling in aged tissues impairs stem cell functionality, while its inhibition demonstrates the capacity to restore tissue regenerative potential[52]. These cross-tissue findings collectively establish the pivotal role of Tgf-β1 in the biology of aging. While our study identifies Tgf-β1 as a key SASP factor driving the osteogenic differentiation of SPC in craniosynostosis, our transcriptomic data also indicate the significant upregulation of other SASP members, such as Vegfa and Cxcl12. These factors likely form a complex synergistic network with Tgf-β1 to collectively exacerbate the pathological process. Future studies are warranted to dissect the precise regulatory interplay among these SASP factors and to determine whether they form a positive feedback loop with Tgf-β1 that irreversibly propels premature suture fusion.

Using in vitro skull cultures[53] and subcutaneous drug injection in mice, we found that interventions such as p38α MAPK and Tgf-β1 inhibition effectively mitigated suture fusion and maintained the quantity of suture mesenchymal components. Although we have confirmed the effectiveness of Tgf-β1 inhibition in alleviating craniosynostosis, histological results showed that it had a negative effect on mineralization and deposition and might damage the integrity of the sutural structure. Another finding of the present study is that cKO of *Mapk14* specifically in Prrx1-expressing cells significantly prevented SPC senescence and ameliorated suture fusion in MUT mice. However, μCT scanning and osteogenic differentiation assays revealed abnormally persistent patency of the postfrontal suture and impaired osteogenic differentiation capacity in SPC from cKO mice. Therefore, we further investigated genetic attenuation therapy using scAAV-shRNA, which significantly mitigated suture fusion and skull deformities by inhibiting the hyperactivation of p38α MAPK in Crouzon mice, and it reduced the elevated ICP in MUT mice and improved behavioral abnormalities. The core rationale for employing shRNA-mediated gene knockdown lies in achieving superior

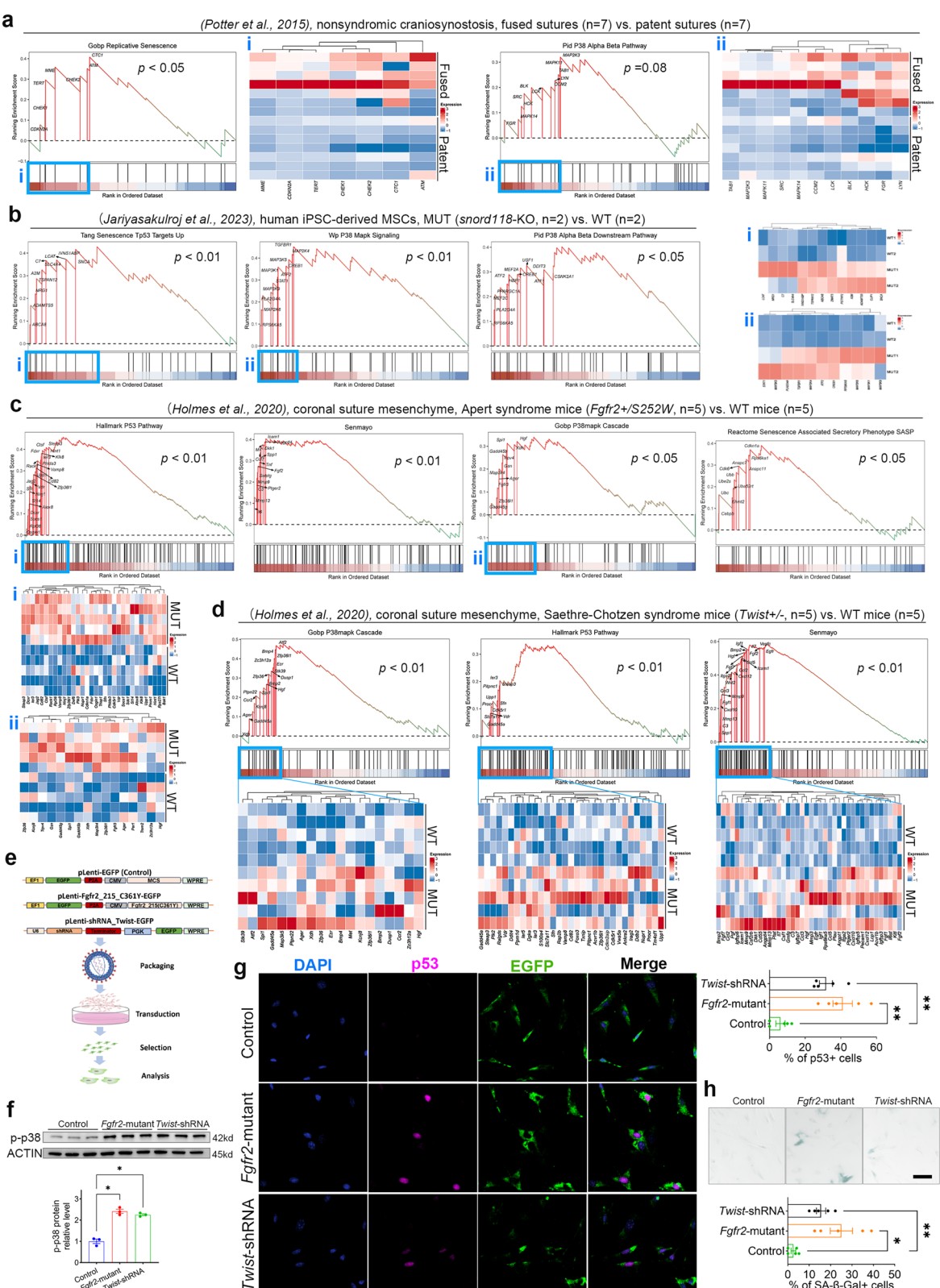

tissue specificity and temporal controllability. Although small-molecule inhibitors offer greater application convenience, they often struggle to avoid off-target effects. In comparison, viral vector-delivered shRNA systems enable partial localization of therapeutic intervention, thereby significantly enhancing treatment safety[54]. More importantly, RNAi-based strategies like shRNA are inherently transient. This transient nature provides ideal conditions for establishing a controllable therapeutic window: short-term intervention during critical progression phases of craniosynostosis can halt pathological processes, while subsequent natural attenuation of vector expression permits recovery of p38 MAPK signaling function during normal development, thereby minimizing long-term impairment of physiological functions.

**Fig. 6 | Activation of p38 and p53, and cellular senescence, are prevalent in cranial suture tissues/cells derived from both human craniosynostosis patients and murine models. a** GSEAs specific to "Replicative Senescence" and "P38 Alpha Beta Pathway" gene sets and heatmaps of core-enriched genes for bulk RNA sequencing data from fused and patent suture tissues of non-syndromic craniosynostosis patients. *N* = 7 per group. **b** GSEAs specific to "Senescence TP53 Targets up", "P38 MAPK Signaling" and "P38 Alpha Beta Downstream Pathway" gene sets and heatmaps of core-enriched genes for bulk RNA sequencing data from WT and MUT (*snord118*−/−) MSCs derived from human iPSC. *N* = 2 per group. **c** GSEAs specific to "P53 Pathway", "Senmayo", "P38 MAPK Cascade", and "SASP" gene sets and heatmaps of core-enriched genes for bulk RNA sequencing data from coronal suture mesenchyme of Apert syndrome mice (*Fgfr2*+/*S252W*) and WT mice. *N* = 5 per group. **d** GSEAs specific to "P38 MAPK Cascade", "P53 Pathway", and "Senmayo" gene sets and heatmaps of core-enriched genes for bulk RNA sequencing data from coronal suture mesenchyme of Saethre-Chotzen syndrome mice (*Twist*+/−) and WT mice. *N* = 5 per group. **e** Schematic diagram of lentiviral plasmid design, packaging, transfection, and subsequent experiments and analysis. **f** Western blotting analysis of p-p38 protein level in MC3T3-E1 cells transfected with lentivirus. *N* = 3. **g** Representative images of immunocyte fluorescent staining with p53 (in red) of MC3T3-E1 cells transfected with lentivirus, and the quantification of p53-positive cells proportion. Nuclei were stained with DAPI (in blue). EGFP (the marker of lentiviral plasmid) was in green. Scale bar: 50 μm. *N* = 5 per group. **c** Representative images of SA-β-Gal staining of MC3T3-E1 cells transfected with lentivirus, and the quantification of positive cells proportion. *N* = 5 per group. All data are presented as mean ± SEM. For **f**–**h**, one-way ANOVA was used, followed by Tukey's test. *\*P* < 0.05, \*\**P* < 0.01.

From another perspective, while our study confirms the significant therapeutic efficacy of targeting p38α MAPK in alleviating pathological craniosynostosis, it is important to acknowledge a critical consideration in its potential translation: how to balance therapeutic benefits against potential impacts on physiological osteogenesis. The observed delayed suture fusion and impaired osteogenic differentiation capacity of SPC in *Mapk14* cKO mice clearly highlight this risk. The p38 MAPK signaling pathway plays a complex dual role in skeletal development and repair: it drives abnormal stem cell senescence and excessive bone formation under pathological conditions, while simultaneously serving as an essential physiological signal for normal bone formation and mineralization[23]. Therefore, complete and sustained inhibition of p38α MAPK may achieve disease treatment at the inevitable expense of impairing physiological growth and remodeling processes. This discovery provides important insights for developing safe therapies targeting senescence or the p38 MAPK pathway in the future. First, future therapeutic strategies may require precise temporal windows—for instance, implementing transient interventions only during active disease progression phases—to avoid long-term effects on skeletal development. Second, exploring lower dosages or more spatially selective administration methods (such as localized drug delivery) could help separate therapeutic effects from side effects. In summary, our research not only reveals a novel therapeutic target but also defines crucial boundary conditions for its safe application.

Cellular senescence is known to profoundly impact tissue microenvironment and stem cell function through the SASP and cell cycle arrest. In bone biology, the accumulation of senescent cells has been shown to disrupt osteoblastic-osteoclastic coupling and impair the osteogenic differentiation potential of mesenchymal stem cells[55]. This aligns with our observation that, while *Mapk14*-cKO rescued senescence and pathological suture fusion, it also led to impaired osteogenic differentiation and inhibited physiological fusion of the suture, underscoring an intricate link between senescence and differentiation programs. Furthermore, while our investigation centers on p38 MAPK, craniosynostosis— particularly forms driven by FGFR mutations—exhibits another hallmark feature: Ras/ERK signaling hyperactivation, which subsequently promotes excessive osteogenic differentiation and extracellular matrix deposition. Both p38 MAPK and Ras/ERK pathways function as central signaling hubs enabling cellular responses to stressors, and both are capable of inducing alterations in cell fate[56]. A well-established synergistic interaction between p38 MAPK and Ras/ERK signaling contributes to hyperosteogenesis in craniosynostosis. Future studies are warranted to delineate how p38 MAPK and Ras/ERK signaling cooperate or act independently to drive the pathological program, and how the convergence of these signaling pathways collectively leads to SPC dysfunction, and ultimately, premature suture fusion.

## Methods
### Animals
In this study, *Fgfr2*^C361Y/+^ mice were engineered to carry the Cys361Tyr point mutation (corresponding to the human Cys342Tyr mutation) based on the *Fgfr2-215* transcript (ENSMUST00000122054.8)[11]. The mouse model was generated by GemPharmatech (Nanjing, China) using CRISPR/Cas9 technology, with synonymous mutations introduced at site 362 to avoid secondary cleavage. For the construction scheme and validation (see Fig. S1b and Table S1). *Mapk14*-KO mice (*Mapk14*^−/+^, Cat. no. NM-KO-225013) and *Mapk14*-Flox mice (*Mapk14*^f/f^, Cat. No. NM-CKO-2102209) were purchased from Southern Model Organisms (Shanghai, China). *Prrx1*-Cre mice (*Prrx1*^cre^, Cat. No. 005584) were purchased from The Jackson Laboratory. *Fgfr2*^C361Y/+^; *Mapk14*^−/+^ mice, *Fgfr2*^C361Y/+^; *Prrx1*^cre/+^; *Mapk14*^f/+^ mice, and *Fgfr2*^C361Y/+^; *Prrx1*^cre/+^; *Mapk14*^f/f^ mice were obtained via hybridization between the individual strains. All mice were maintained on a C57BL/6J background throughout the study period. All animal experiments and procedures were conducted in accordance with the National Institutes of Health Guidelines for the Care and Use of Laboratory Animals and were approved by the Plastic Surgery Hospital, Chinese Academy of Medical Sciences, and Peking Union Medical College (No. 2023-A-46). All animal experiments were conducted in accordance with the relevant institutional guidelines and regulations.

### Cell preparation
Isolation of mouse cranial suture cells and subsequent experiments were performed according to a previously described protocol[57]. Briefly, the coronal suture tissue was micro-dissected, minced, and incubated in DPBS (Gibco) containing 0.2% collagenase A (Roche), 10 mM HEPES (Gibco), 1% penicillin–streptomycin (PS, Gibco), and 2% fetal bovine serum (FBS, VivaCell) at 37°C for 1 h on a shaker. The digested solution was filtered through a 40 μm nylon mesh, followed by centrifugation at 400×*g* for 7 minutes at 4 °C. The resulting cell pellets were resuspended in DMEM containing 15% FBS for 2–4 h to restore cell viability and remove debris. Subsequently, the cells were digested with Accutase (Gibco) for flow cytometric analysis or FACS detailed below.

### Drug administration
The following drugs were administered subcutaneously once daily into the scalp of mice: SB203580 (MedChemExpress [MCE]) at 3 or 6 mg/kg; pifithrin-α (MCE) at 2 or 4 mg/kg; and SB525334 (MCE) at 5 or 10 mg/kg. All drugs were prepared as stock solutions in dimethyl sulfoxide (DMSO; MCE). Control mice received control vehicle injections. Treatments were randomly assigned, and injections continued for 14 days (until P14). Coronal suture tissues were collected from P14 and P28 mice for histological analysis.

### Injection of scAAVs
scAAV9-shRNA_*Mapk14*-EGFP and scAAV9-shRNA_Scramble-EGFP were designed using the VectorBuilder platform (Guangzhou, China) and constructed by OBiO Technology (Shanghai, China). The target transcripts for shRNA are NM_001168508.1, NM_001168513.1, NM_001168514.1, and NM_011951.3, targeting their coding sequences (CDS) region. scAAVs were injected under the scalp of newborn mice at a dose of 10^10 GC/g. Seven days post-injection, samples were collected to assess the expression of shRNA (EGFP). Samples from 1M mice were collected for histological staining. The vector information of pscAAV9 was shown in Table S1.

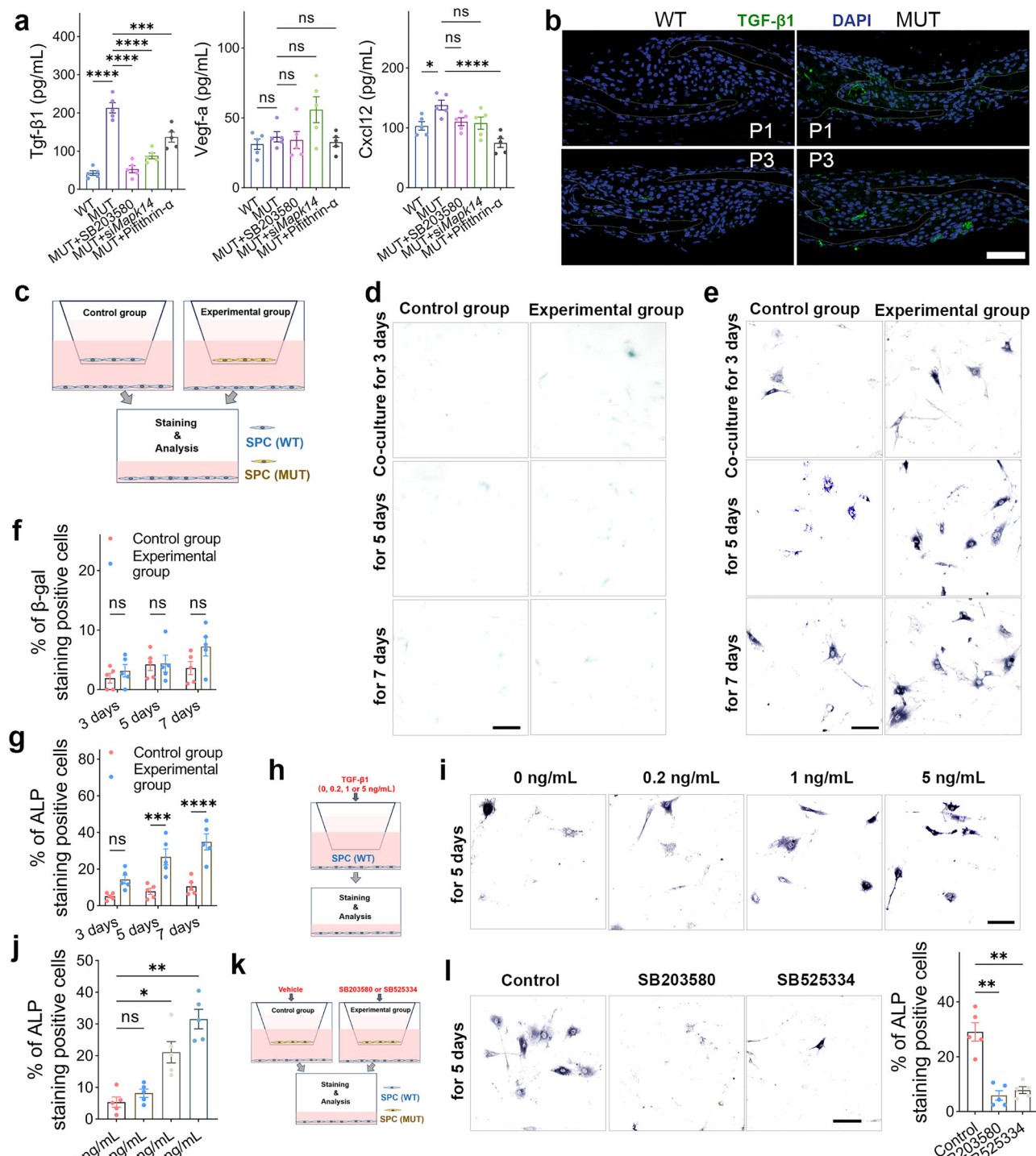

**Fig. 7 | The Tgf-β1 secreted by senescent cells affects the osteogenic differentiation of SPC. a** Quantification of the key SASPs protein level in SPC culture supernatant with/without inhibitors/siRNA treatment. $N = 5$ per group. **b**, **c** Representative immunofluorescence images of Tgf-β1 in coronal sutures of WT and MUT mice (P1 and P3). Scale bar: 50 μm. **c** Schematic diagram of SPC co-culture. Representative images (**d**) of SA-β-Gal staining in co-cultured SPC and the quantification (**f**) of positive cells. Scale bar: 50 μm. $N = 5$ per group. Representative images (**e**) of ALP staining in SPC after co-culture and the quantification (**g**) of positive cells. Scale bar: 50 μm. $N = 5$ per group. **h** Schematic diagram of gradient concentration Tgf-β1 protein treatment and staining of SPC. Representative images **i** of ALP staining in SPC with/without treatments of Tgf-β1 and quantification **j** of the positive cells. Scale bar: 50 μm. $N = 5$ per group. **k** Schematic diagram of SPC co-culture with/without inhibitor treatment. **l** Representative images of ALP staining of co-cultured SPC with/without inhibitor treatment and quantification of positive cells. Scale bar: 50 μm. $N = 5$ per group. All data are presented as mean ± SEM. For **a**, **j**, and **l**, one-way ANOVA was used, followed by Dunnett's test. For **f** and **g**, two-way ANOVA was used, followed by Sidak's test. *$P < 0.05$, **$P < 0.01$, ***$P < 0.001$, ****$P < 0.0001$. ns, not significant.

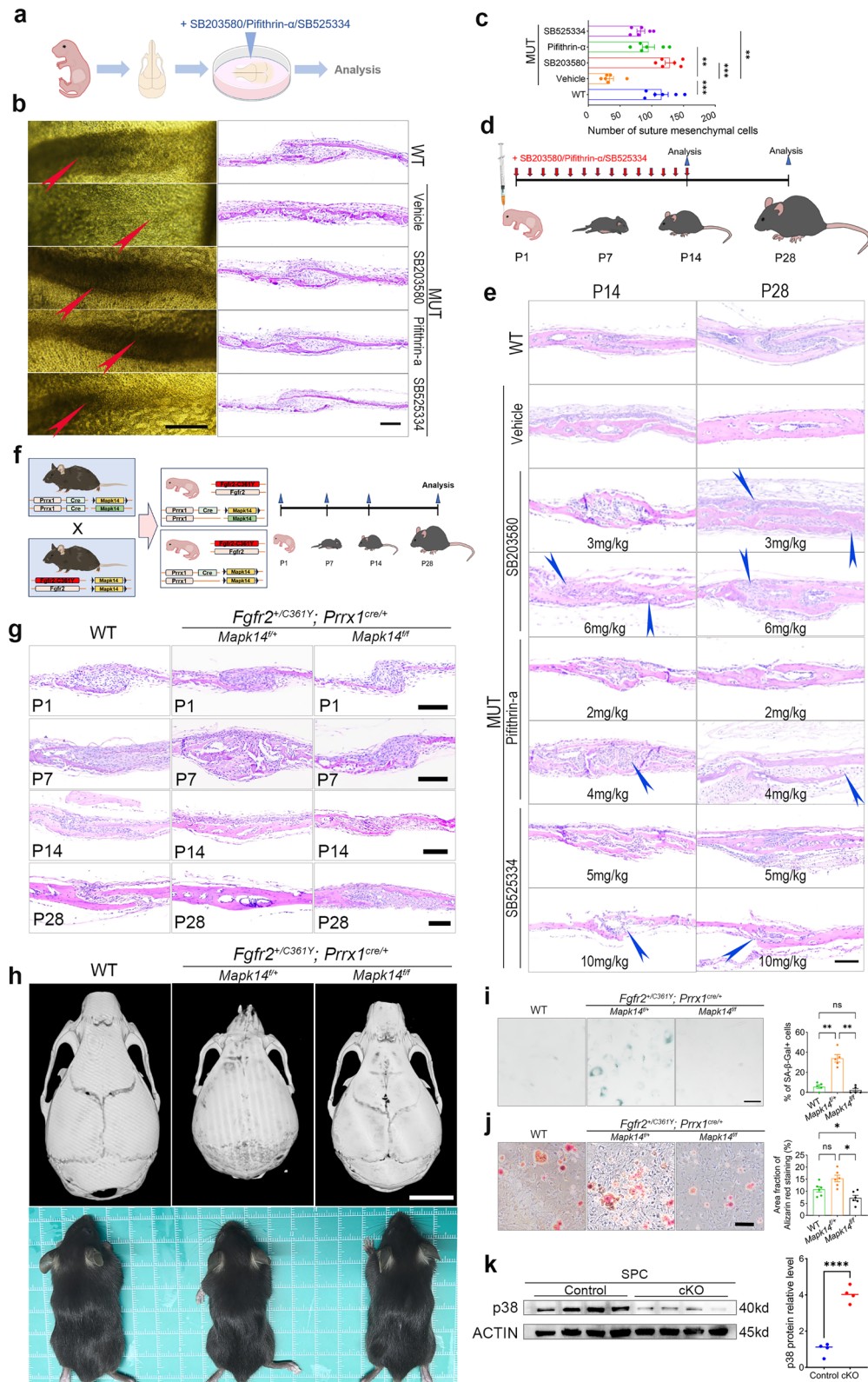

## µCT scanning and analysis

Cranial imaging of mice at different ages was performed using the Quantum FX µCT imaging system (PerkinElmer, USA) with a resolution of 20 µm. Reconstruction and analysis were conducted using the 3D Slicer software (v5.6.2) and the PerkinElmer analysis system.

## Intracranial pressure (ICP) measurement

The ICP measurement method for mice refers to the description of previous studies[58]. In brief, after anesthetizing mice with isoflurane, heat support was provided through a warm water pad to ensure that the body temperature of the anesthetized mice remained constant. Made a sagittal incision on the

**Fig. 8 | Pharmacological and genetic attenuation therapy of p38α MAPK mitigates SPC senescence, suture fusion or/and skull deformities in MUT mice.**
**a** Schematic diagram of the calvaria tissue explants model. **b** Representative images of coronal sutures of the explants treated with inhibitors and H&E staining. The red arrow indicated the overall appearance of the coronal sutures of explants. Scale bar (left): 500 μm. Scale bar (right): 100 μm. **c** Quantification of mesenchymal cells number in sutures of explants treated with inhibitors. $N = 6$ explants/group.
**d** Schematic diagram of inhibitors injection treatment and the following analysis. **e** H&E staining images of coronal sutures in mice with/without regular inhibitor treatment. The blue arrow indicated the relief of suture fusion. Scare bar: 100 μm. $N = 3–5$ mice/group. **f** Schematic diagram of construction and subsequent analysis for $Fgfr2^{C361Y/+}$; $Prrx1^{cre/+}$; $Mapk14^{f/f}$ mice ($Mapk14^{f/f}$ cKO mice) and $Fgfr2^{C361Y/+}$; $Prrx1^{cre/+}$; $Mapk14^{f/+}$ mice ($Mapk14^{f/+}$ cKO mice). **g** Representative images H&E staining of coronal sutures from WT mice, $Mapk14^{f/+}$ cKO mice, and $Mapk14^{f/f}$ cKO

mice at P1, P7, P14, and P28. Scale bar: 100 μm. **h** The overall appearance images and μCT scanning images of skulls from WT mice, $Mapk14^{f/+}$ cKO mice, and $Mapk14^{f/f}$ cKO mice at P28. Scale bar: 5 mm. **i** Representative images of SA-β-Gal staining and quantification of positive cells proportion of SPC from WT mice, $Mapk14^{f/+}$ cKO mice, and $Mapk14^{f/f}$ cKO mice. Scale bar: 100 μm. $N = 5$/group. **j** Representative images of Alizarin red staining and quantification of staining area fraction of SPC from WT mice, $Mapk14^{f/+}$ cKO mice, and $Mapk14^{f/f}$ cKO mice. Scale bar: 100 μm. $N = 6$/group. **k** Western blotting analysis of p38 protein in SPC of WT mice (Control group) compared to $Fgfr2^{C361Y/+}$; $Prrx1^{cre/+}$; $Mapk14^{f/f}$ cKO mice, and the relative quantitative result of p38 protein. cKO, conditional knockout. $N = 4$. All data are presented as mean ± SEM. For **k**, two-tailed $t$-test was used. For **c**, **i**, and **j**, one-way ANOVA was used, followed by Dunnett's test. *$P < 0.05$, **$P < 0.01$, ***$P < 0.001$, ****$P < 0.0001$. ns, not significant.

---

scalp of mice. The probe connected to the intracranial pressure monitoring system (FISO, Canada) penetrated the left parietal bone (~2 mm outside the sagittal suture and 2 mm before the lambdoid suture). Inserted the needle to the calculated depth without applying external pressure to the skull, positioned it in the subarachnoid space, fixed the probe, and recorded the ICP once it stabilized or began to decrease.

### Behavioral assays
Novel object test: During habituation, mice were placed in a test chamber ($30 \times 50 \times 25$ cm³) for 5 min. They then explored two identical objects for 10 min during familiarization. In the test session, mice were presented with one familiar and one novel object of different shapes (similar size) and given 5 min to explore. The preference index was calculated as $(T_n–T_f)/(T_n + T_f) \times 100\%$ where $T_n(T_f)$ represents the time spent exploring new (familiar) objects.

Rotarod test: In the initial training stage, mice were trained to stay on a rotating rod at a constant 4 rpm speed for 30 s. During the test stage, the bar speed accelerated from 4 to 40 rpm over 5 min, and the time until the mice fell was recorded. Each mouse was tested three times with 15 min intervals.

### Flow cytometry
Flow cytometry was performed using a BD FACSCelesta for analysis or a BD FACSAria III for sorting. The experimental protocols and gating strategies were adapted from previous studies[8,19]. Single-cell suspensions were prepared and washed 1–2 times with pre-cooled staining buffer (PBS containing 2% FBS and 1 mM EDTA), followed by blocking with Fc blocking buffer (Elabscience; 1:50) at 4 °C for 20 min DAPI staining solution (Thermo Fisher Scientific, 0.1 μg/100 μl) was added, and the cells were incubated in the dark for 5 min. After washing twice with flow buffer, cells were incubated with the secondary antibody at 4 °C for 20 min. After washing, the cells were analyzed or sorted. The SPC population was defined as CD45⁻Ter119⁻Tie2⁻6C3⁻Thy1⁻CD105⁻CD51⁺CD200⁺. The SPC ratio was determined as the proportion of CD51+ CD200+ cells within the single-cell population after de-adhesion and removal of DAPI+ cells. The FlowJo software (v10.8.1) was used for data analyses. FACS was used to enrich the SPC from the coronal suture tissues, and the sorted cells were used for further culturing and experiments. The detailed information on antibodies could be found in Table S1.

### Cell cycle analysis
SPCs were collected and fixed in pre-cooled 70% ethanol at 4 °C for at least 2 h. After washing, RNase A (Thermo Fisher Scientific, 5 μg/100 μl) was added, and the DNA was labeled with 7-aminoactinomycin D staining solution (BD, 0.25 μg/100 μl). The cells were analyzed using a BD FACSCelesta to determine the distribution of cells across different cell cycle phases.

### Cell culture
SPCs were cultured in DMEM containing 10% FBS and 1% PS in a 37 °C incubator with 5% $CO_2$. SB203580 (10 μM) and pifithrin-α (5 μM),

SB525334 (1 μM), and control (DMSO, 0.1%) were used to treat the cells. siRNA transfections (Santa Cruz Biotechnology) followed the manufacturer's protocol, using control siRNA, Mapk14 siRNA, and Mapk11 siRNA at a working concentration of 60 nM for 48 h. Post-treatment, the cells were collected for downstream analysis or staining. The key reagents or assay kits for cell separation, culture, and experiments could be found in Table S1.

### Cell staining
SPCs were plated at appropriate densities, and the Senescence β-Galactosidase Staining Kit (Cell Signaling Technology [CST]) and ALP Staining Kit (Beyotime) were used according to the manufacturer's instructions. Osteogenic differentiation was induced for 3–7 days prior to ALP staining, as per the manufacturer's instructions (Oricell).

Osteogenic, chondrogenic, and chondrogenic differentiation of SPC (Oricell) was carried out and stained following the manufacturer's protocols.

### Comet assay
After the SPCs were sorted and lysed, their DNA was unwound, electrophoresed, and stained using a Comet Assay Kit (Abbkine). Samples were observed using a Leica SP8 confocal imaging system. Data processing was performed using the OpenComet software (v1.3).

### 5-ethynyl-2′-deoxyuridine (EdU) cell proliferation assay
After cell plating, EdU (10 μM) was added and co-cultured for 2 h. Cells were fixed, permeabilized, and subjected to fluorescence labeling and DNA staining. using the E-Click EdU Cell Proliferation Imaging Assay Kit (Elabscience) following the manufacturer's instructions. Observations were made using a Leica SP8 confocal imaging system.

### Transfection of lentiviral plasmids into MC3T3-E1 cells
pLenti-shRNA_*Twist*-EGFP was designed using the VectorBuilder platform (Guangzhou, China) and constructed by OBiO Technology (Shanghai, China). The target transcript for shRNA is NM_011658.2, targeting its 3 'UTR region. Vector information of lentiviral plasmid was shown in the KEY RESOURCES TABLE and Fig. S5c. The design and construction of lentiviral plasmids for pLenti-*Fgfr2*_215-C361Y-EGFP (C361 mutation corresponds to human *FGFR2-C342Y* mutation) and pLenti-EGFP (as a control) have been described in detail in our previous research[33]. The transfection procedure was as follows: MC3T3-E1 cells (in the logarithmic growth phase) were collected. The lentivirus was diluted to a multiplicity of infection (MOI) of 20 and supplemented (without polybrene) for cell transfection. After 24 h of viral exposure, the virus-containing medium was removed and replaced with fresh culture medium. The cells were subsequently maintained for an additional 24–48 h before puromycin (HANBIO, 3 μg/ml) or ampicillin (Beyotime, 70 μg/ml) resistance selection. After more than 3 days of drug selection, the cells were collected for subsequent experiments and analysis.

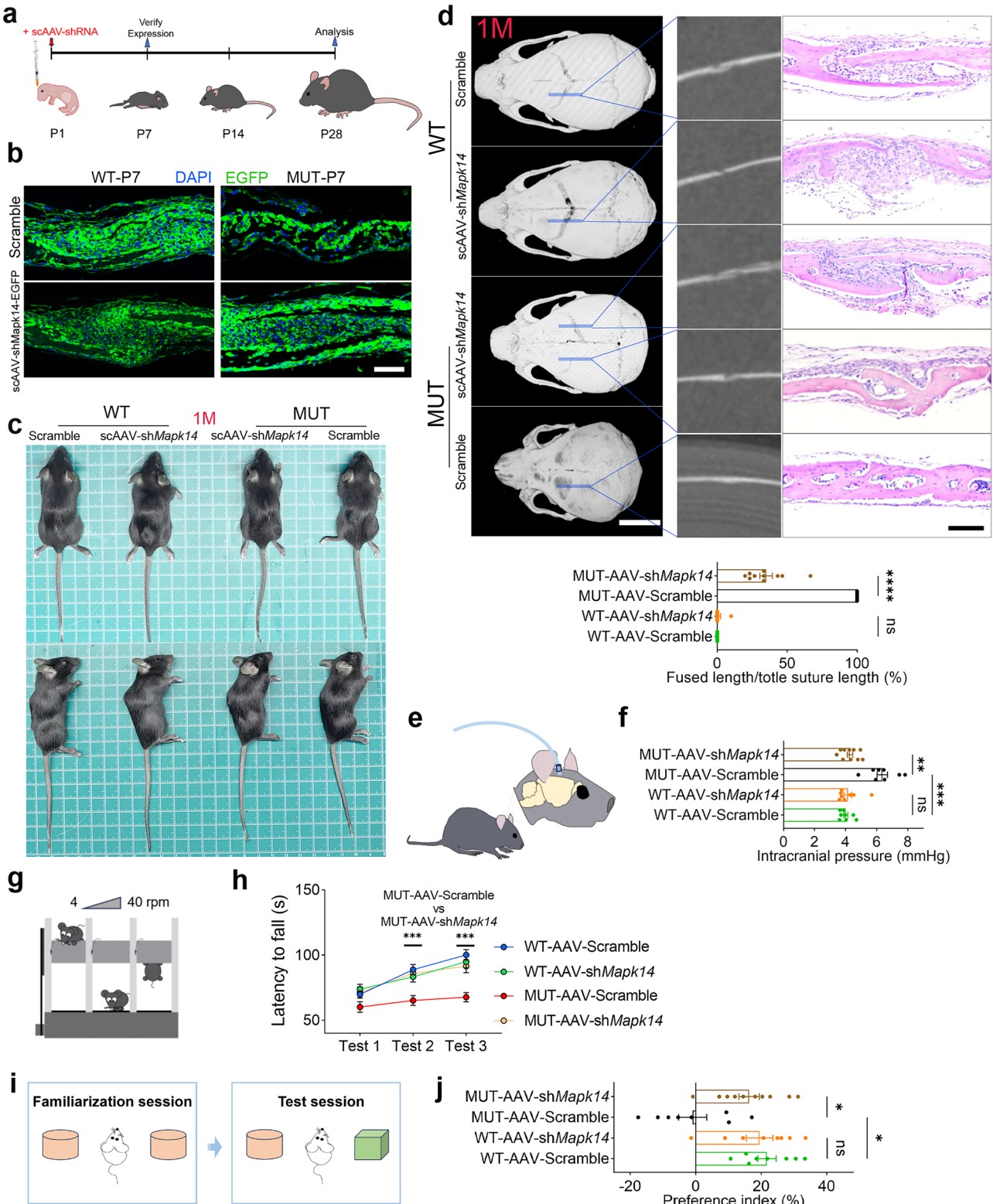

**Fig. 9 | The scAAV-sh*Mapk14* attenuation therapy significantly alleviates Crouzon craniosynostosis. a** Schematic diagram of scAAV injection and subsequent analysis. **b** The expression of EGFP (scAAV-EGFP) in coronal sutures of mice (P7) with scAAV injection. *N* = 3 mice/group. **c** The overall appearance images of WT and MUT mice with scAAV injection at 1M. **d** The µCT scanning images and H&E staining images of WT and MUT mice with scAAV injection, and quantification of fused length/total coronal suture length percentage of WT and MUT mice with scAAV injection. Scale bar for µCT scanning images: 5 mm. Scale bar for H&E staining images: 100 µm. *N* = 10 mice in the MUT-AAV-sh*Mapk14* group and 8 mice in the other groups. **e** The schematic of the intracranial pressure (ICP)

measurement setup. **f** Quantification of ICP of mice with scAAV therapy. *N* = 10 mice in the MUT-AAV-sh*Mapk14* group and 8 mice in other groups. **g** The schematic of the rotarod test. **h** Rotarod performance and quantification of the latency to fall. *N* = 10 mice in MUT-AAV-sh*Mapk14* group and 8 mice in the other groups. **i** The schematic of the novel object test. **j** Quantification of the preference index of novel object. *N* = 10 mice in the MUT-AAV-sh*Mapk14* group and 8 mice in other groups. All data are presented as mean ± SEM. For **c**, **f**, and **j**, one-way ANOVA was used, followed by Tukey's test. For **h**, two-way ANOVA was used, followed by Dunnett's test. *$P < 0.05$, **$P < 0.01$, ***$P < 0.001$, ****$P < 0.0001$. ns, not significant.

## siRNA sub-library screening

The p53 activation-related siRNA sub-library screening system was designed based on genes involved in p53 activation and regulation[22]. RiboBio Biotech (Guangzhou, China) customized the system. SPCs were seeded into a 96-well plate and transfected with the siRNA system alongside pp53-TA-luc (Beyotime, 0.25 μg/100 μl) and pRL-TK (Beyotime, 0.1 μg/100 μl) using Lipofectamine 2000 (Invitrogen, 0.8 μg/100 μl) as the transfection reagent. After 48 h, a Luciferase Reporter Gene Assay Kit (Beyotime) and multifunctional microplate reader (BioTek, USA) were used to measure the expression levels of Firefly and Renilla luciferases. The ratio of these levels was calculated to determine the p53 activation level in each SPC group.

## Histology

Skull or cranial suture specimens were collected and fixed in 4% paraformaldehyde (PFA, Beyotime) overnight at 4 °C. Depending on the age of the mice, decalcification was performed in EDTA (Biosharp) for 0–30 days at 4 °C. Paraffin or OTC frozen embedding medium (Sakura Finetek) was used as appropriate for the experimental requirements. Paraffin sections were cut to a thickness of 5 μm, whereas frozen sections were cut to 10 μm. H&E staining of the sections followed the manufacturer's instructions (Beyotime).

## Tissue SA-β-Gal staining

Tissue sections were hydrated, permeabilized, and stained using the senescence β-galactosidase staining kit (CST) according to the manufacturer's protocol.

## Immunohistochemical staining

Paraffin sections were dewaxed, hydrated, and subjected to antigen retrieval using sodium citrate solution under heat. After cooling, sections were washed twice with PBS, permeabilized with PBS containing 0.3% Triton X-100 (Beyotime) for 15 min, and blocked with PBS containing 5% goat serum for 30 min. Primary antibodies were diluted in antibody diluent (Beyotime), and sections were incubated overnight at 4 °C. After washing three times with PBS, endogenous peroxidase activity was blocked using a 3% hydrogen peroxide solution, followed by an additional PBS wash. Secondary antibody incubation and color development were performed using a two-step immunohistochemical staining kit (Zsbio). Hematoxylin nuclear counterstaining was performed as described above. The detailed information of antibody could be found in Table S1.

## Immunofluorescence staining

Tissues: The sections were thawed at room temperature, hydrated with PBS, permeabilized with PBS containing 0.3% Triton X-100 for 15 min, and blocked with PBS containing 5% goat serum (Beyotime) for 30 min. Primary antibodies were diluted in antibody diluent, incubated overnight at 4 °C, and washed three times with PBS. Secondary antibodies were diluted, incubated at room temperature for 1 h, and washed three times with PBS. Finally, the slides were mounted using the Antifade Mounting Medium with DAPI (Abcam).

Cells: Once the cells reached an appropriate density, they were washed twice with pre-cooled PBS, fixed in 4% PFA for 15 min, and washed again with PBS. Permeabilization was performed with PBS containing 0.3% Triton X-100 for 15 min, followed by blocking with PBS containing 5% goat serum for 30 minutes. Primary antibodies were diluted in antibody diluent and incubated with the cells overnight at 4 °C. After washing three times with PBS, secondary antibodies were diluted and incubated with the cells at room temperature for 1 h, and washed three times with PBS. Finally, the slides were mounted using the Antifade Mounting Medium with DAPI and imaged.

Imaging was performed using a Leica SP8 confocal imaging system, and data processing was performed using the LAS X software (v3.7.5). The detailed information on antibodies could be found in Table S1.

## Enzyme-linked immunosorbent assays (ELISA)

After centrifugation of each group of cell culture media, the supernatant was collected, and the protein concentrations of Tgf-β1, Vegf-a, and Cxcl2 in the supernatant were quantified using ELISA Kits (Beyotime). Tgf-β1 mostly exists in the form of inactive complexes and must be activated with hydrochloric acid solution before detection. According to the instructions of the reagent supplier, perform the detection procedure. After sample addition, incubation, antibody binding, and color reaction, terminated the reaction, and quickly measured the absorbance at A450 nm. Calculate the corresponding protein concentration based on the standard curve.

## Western blotting

Tissues or cells were lysed in lysis buffer (Beyotime) to extract total protein, and protein concentrations were measured using a protein concentration assay kit (Beyotime). Samples were loaded and separated on a 12% Bis–Tris polyacrylamide gel. The separated proteins were transferred to a polyvinylidene fluoride membrane (Millipore), blocked with 5% skim milk, and incubated with primary antibodies overnight at 4 °C. After incubation with secondary antibodies for 1 h, signals were detected using the ECL Plus reagent (Beyotime). The ImageJ software (Fiji package; v1.54) was used to analyze the protein bands. The detailed information on antibodies could be found in Table S1.

## Quantitative real-time polymerase chain reaction (q-PCR)

Total RNA was isolated from SPC using the TRIzol method (Invitrogen). RNA was reverse transcribed using a HiScript III 1st Strand cDNA Synthesis Kit (Vazyme). Gene expression profile was analyzed by CFX Connect (Bio-Rad, USA) using Taq Pro Universal SYBR qPCR Master Mix (Vazyme). The relative mRNA level in each sample was normalized to its *Actb* content. Values are provided as relative to *Actb* expression. The primer sequence could be found in Table S1.

## Calvarium explant culture

Postnatal mouse calvarial tissue was microdissected according to a previously established protocol[8,53], ensuring the intact preservation of major cranial sutures, periosteum, and dura mater. After washing twice with PBS, the calvariae were cultured in DMEM containing 10% FBS and 1% PS in 24-well plates. The culture medium was changed every 2 or 3 days. Drug treatments were added according to the treatment plan: SB203580 hydrochloride, pifithrin-α hydrobromide, SB525334, and control (DMSO). After 10 days of culture, the tissues were fixed with 4% PFA at 4 °C for 24 h before embedding and histological staining.

## Bulk RNA sequencing and analysis

Coronal suture tissues were dissected from P3 mice, and total RNA was extracted using TRIzol Reagent. RNA quality was verified by using Bioanalyzer 2100 (Agilent, USA). RNA library and sequencing were performed by Novogene Technology Company (Beijing, China). Differential expression analysis between the two groups was performed using the DESeq2 R package, and enrichment analysis and plotting were performed using the clusterProfiler package.

Bulk RNA sequencing data from public datasets[27–31] were also analyzed by R software for enrichment analysis and plotting. In brief, RNA sequencing data derived from cranial suture tissues or cells of human and murine craniosynostosis series were analyzed through GSEA using gene sets including "p38 MAPK pathway", "p53 pathway", and "cellular senescence" (obtained from the GSEA platform, while "Senmayo" was a new set of genes described in previous studies[32] that identifies senescent cells and predicts senescence-associated pathways across tissue). Heatmaps were subsequently generated to visualize the expression patterns of core-enrichment genes[59].

## Statistics and reproducibility

All data are presented as mean ± SEM. For comparisons between two groups, the unpaired two-tailed Student's *t*-test was used. For comparisons

of three or more groups, one-way or two-way ANOVA was used if the normality test passed, followed by Sidak's/Tukey's/Dunnett's multiple comparison test for all groups, unless otherwise stated. Statistical analysis was performed using GraphPad Prism (v9.5.0). $P < 0.05$ was considered statistically significant. *$P < 0.05$, **$P < 0.01$, ***$P < 0.001$, ****$P < 0.0001$. ns, not significant. Alphabet method was also used to indicate significance. Values for all data points in graphs are reported in the Supporting Data 1 file.

## Reporting summary

Further information on research design is available in the Nature Portfolio Reporting Summary linked to this article.

## Data availability

Bulk RNA-seq have been deposited at Genome Sequence Archive (GSA in National Genomics Data Center, https://ngdc.cncb.ac.cn/gsa/) at GSA: CRA018981 and are publicly available as of the date of publication. Source data underlying all manuscripts can be found in the Supplementary Data 1 file. The unedited blot images are displayed in the Supplementary Information.

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

## Acknowledgements

This study was supported by the Beijing Natural Science Foundation (Grant No. 7242121) and the National Natural Science Foundation of China (Grant No. 82402947). Graphical Abstract Image was created with BioGDP.com (Agreement number: GDP2025NWQE27). Portions of the schematic diagrams in Figs. 3, 6, 8, and 9 were adapted from SciDraw.io. We gratefully acknowledge the SciDraw illustration library and the contributing authors (Diogo Losch De Oliveira, https://doi.org/10.5281/zenodo.6808878; Nicolás De Francesco, https://doi.org/10.5281/zenodo.4912419; Diogo Losch De Oliveira, https://doi.org/10.5281/zenodo.6808886; Annie Park, https://doi.org/10.5281/zenodo.10940481; Health Robinson, https://doi.org/10.5281/zenodo.7058520; Health Robinson, https://doi.org/10.5281/zenodo.7058534; Elisa Galliano, https://doi.org/10.5281/zenodo.3926410) for their contributions to this study.

## Author contributions

Conceptualization: X.L.J. and Z.C.; performing experiments: Z.C., Z.Y.C., X.Y.C., Y.Y.Y., Y.W., X.Y.H., X.S.G., C.Z.L., and G.D.S.; writing—original draft: Z.C. and Z.Y.C.; writing—review and editing: Z.C., Z.Y.C., and X.L.J.; funding acquisition: X.L.J.; resources: X.L.J.; and supervision: X.L.J.

## Competing interests

The authors declare no competing interests.
