## [Transparent Peer Review file · Communications Biology]

P38 α MAPK-Induced Senescence in Cranial Suture Progenitor Cells Promotes Craniosynostosis

Corresponding Author: Professor Xiaolei Jin

Version 0:

Reviewer comments:

Reviewer #1

(Remarks to the Author)

The article entitled "P38 α MAPK-Induced Senescence in Cranial Suture Progenitor Cells Promotes Craniosynostosis" by Zong Chen et al. is a comprehensive effort to elucidate the molecular mechanism in a type of the Crouzon craniosynostosis induced by the C361Y mutation of the Fgfr2 receptor. The authors show that cellular senescence induced by p38/p53 activation promoted osteogenic differentiation via paracrine Tgf- β 1. They further suggest that p38 inhibition could ameliorate the synostotic phenotype making it a potential therapeutic target.

The experiments are well planned, performed, analyzed and interpreted, supporting the authors' conclusions. The questions are addressed by multiple and diverse approaches validating the given answers.

One important question that does not appear to be of concern to the authors, is the role of Erk1/2 kinases in the Crouzon syndrome that have been shown to mediate the downstream effects. The interplay of Erk/p38/Tgf-beta in several systems including osteogenesis, has previously been shown and the possibility that the observed effects may be affected and/or mediated by Erks cannot be excluded. Validating if the observed role of p38/p53/Tgf-beta on senescence and differentiation are Erk-independent effects or not, would greatly improve the manuscript.

There are few other points that could be improved and clarified.

1) Regarding the analysis of senescence-related proteins (L118-128, Fig 1g-1, Fig S2) it is not clear which cells were isolated and how was verified the absence of bone cells. The cited reference #49 makes clear that is an easy to mishandle process. Some expression data should be provided to indicate the absence or at least the comparable abundance of non-mesenchymal stem cells in the samples.

2) The quantitative results in several plots are given as "area fraction" or "absolute number" (in a suture). Given the differences in the suture size the absolute number or the area fraction do not necessarily reflect the quantitative differences in the state of the suture cells and can easily be over or under evaluated. Percentage of suture mesenchymal cells should be used throughout.

3) In figure 3b it would appear the CD200+ population in the mutant animals remains clearly distinct from the CD200- suggesting the maintenance of their original characteristics from earlier time points, albeit the overall decrease in cell numbers which would be hard to consolidate with the apparent change in cell cycle in figure 3d. Is the cell cycle difference observed in all time points?

4) The large osteo nodules in figure S4b in the mutant maybe indicative of an increased earlier proliferation state followed by the differentiation state. Is the kinetics of ALP and alizarin red staining in WT and mutant similar to the endpoint shown?

5) Is hard to tell with the 3-color images of figure 3m the p21 positive cells appear to be much more than 8-12% shown in figure 2c. Is the variation so large? Maybe single (B&W) images for each channel would be more informative.

6) The findings indicate that Tgf-beta may be responsible for the phenotype. Is Tgf-beta1 treatment alone, sufficient to induce synostosis in calvaria explant cultures?

The above points should better elucidate the processes affected and the relevant time point.

Finally, given the focus of the manuscript in the role of senescence in craniosynostosis, a small paragraph or few sentences could be added in the introduction or the discussion addressing the role of senescence in bone mesenchymal differentiation and how this relates to p38 but also Ras/Erk overactivation.

Reviewer #2

(Remarks to the Author)

This work investigates the role of p38 α MAPK activation in cranial suture progenitor cells (SPCs) during craniosynostosis. Using a combination of genetically engineered mouse models, patient-derived datasets, in vitro assays, pharmacological and genetic interventions, the authors show that hyperactivation of p38 α MAPK drives SPC senescence via the p38–p53–p21 axis. Senescent SPCs secrete TGF- β 1, enhancing osteogenic differentiation of neighboring SPCs and leading to premature suture fusion. Both pharmacological inhibition and genetic attenuation, Mapk14 knockout or scAAV-shRNA, mitigate SPC senescence, rescue suture patency, and improve craniofacial and behavioral outcomes. This work provides strong mechanistic evidence linking SPC senescence with craniosynostosis, advancing understanding of cranial suture biology and disease pathogenesis.

Major Comments

1. This work convincingly demonstrates a role for senescence in craniosynostosis. However, the discussion section needs to be enriched to increase this work's impact further. I recommend citing and discussing the following highly relevant references: PMID: 35485439 (MSC senescence and combating strategies), PMID: 39303104 (spatial distribution of Gli1+ SPCs in craniofacial region), 37171117 (SPCs and craniofacial development), PMID: 37681346 (inflammaging of SPC niche), PMID: 39486404 (a concise overview of SPCs).
2. Although inhibition of p38 α MAPK is effective, the discussion should more carefully address the risks of impairing physiological osteogenesis, as seen in the Mapk14 cKO mice with delayed mineralization (or please present more related data if possible). Long-term effects and safety considerations for translational therapies should be emphasized.
3. This work highlights TGF- β 1 as a key SASP factor mediating osteogenic differentiation. However, other SASPs, for example, IL-6, VEGFA, and CXCL12, were upregulated at least at the transcription level. Expanding the discussion on whether these could synergize with TGF- β 1 would strengthen the conclusions. Moreover, TGF- β 1 has been found to be implicated in other types of somatic stem cell senescence, for example, PMID: 39743213 (TGF- β 1-induced periodontium stem cell senescence), PMID: 31959867 (TGF- β 1-induced AT2 cell [stem cell in lung] senescence). Please expand the discussion of the effect of TGF- β 1 in a broader scope and/or context.

Minor Comments

1. Please double-check all abbreviation usage for consistency.
2. Some sentences in the manuscript are too long; please consider breaking them into shorter statements to improve clarity and delivery.
3. Figures S6 and S7 contain critical mechanistic evidence; please consider moving parts of them into the main figures.

Recommendation

This is a high-quality and innovative work that significantly advances the mechanistic understanding of craniosynostosis. With improved contextualization in the relevant literature, expanded discussion of safety and translation, and minor editorial refinements, it will be well-suited for publication in Communications Biology.

Reviewer #3

(Remarks to the Author)

The manuscript addresses a relevant and timely aspect of craniosynostosis pathophysiology by identifying the role of the senescence process as a central mechanism driving the abnormal behavior of mesenchymal stromal cells within the cranial suture niche. It also explores a potential therapeutic target, thereby offering alternative strategies for treatment. The work is clearly written and of interest to the field. Only minor revisions are required to further strengthen the study.

Specific comments:

1. The Discussion could be strengthened by including an overview of therapeutic strategies currently explored in vitro, with particular attention to those targeting FGFR2 variants (e.g., DOI: 10.1016/j.omtn.2024.102427).
2. The authors are encouraged to elaborate in the Introduction or Discussion on the rationale for selecting shRNA over a pharmacological approach, especially in view of the therapeutic window supported by transient RNAi-based molecules.
3. The Results section does not consider recent studies showing that DDR2+ mesenchymal stem cell subpopulations within cranial sutures represent primary drivers of pathological premature ossification in craniosynostosis (DOI: 10.1038/s41586-023-06526-2). This aspect should be addressed and its implications for the findings discussed.
4. The authors should evaluate whether specifically targeting the mutant allele of the disease-causing gene in gain-of-function conditions (e.g. Crouzon syndrome) could restore cell cycle progression, reduce the senescence phenotype, and minimize off-target effects.
5. The potential application of functionalized nanoparticles for cell-specific targeting should also be evaluated, or at least discussed, as this approach could provide additional translational relevance
6. Consistency in the use of past and present tense throughout the manuscript should be ensured.

7. The representative images of ALP staining (e.g. figures 5g-5h, 6h, 7d-l etc) should be replaced with higher-quality ones.
8. The Discussion should include a more detailed explanation of p38 signaling, clarifying its role as a downstream effector of FGFR2 or its relationship with Twist pathways.

Version 1:

Reviewer comments:

Reviewer #1

(Remarks to the Author)

The authors address my concerns adequately.

Reviewer #3

(Remarks to the Author)

The revised version of the manuscript shows substantial improvement and can be considered for publication. It presents innovative aspects in the study of craniosynostosis and is supported by a solid and extensive set of experimental data.

Responds to the reviewer's comments:

Reviewer #1:

We feel great thanks for your professional review work on our article. As you are concerned, there are several problems that need to be addressed. According to your nice suggestions, we have made extensive corrections to our previous draft, the detailed corrections are listed below.

1. One important question that does not appear to be of concern to the authors, is the role of Erk1/2 kinases in the Crouzon syndrome that have been shown to mediate the downstream effects. The interplay of Erk/p38/Tgf-beta in several systems including osteogenesis, has previously been shown and the possibility that the observed effects may be affected and/or mediated by Erks cannot be excluded. Validating if the observed role of p38/p53/Tgf-beta on senescence and differentiation are Erk-independent effects or not, would greatly improve the manuscript.

The author's answer:

Thank you for your constructive comments. We agree with the helpful suggestions, and we also conducted experimental studies on the effect of Erk1/2 activation on SPC cell senescence. We treated SPC with gradient concentrations of an Erk1/2-specific inhibitor (SCH772984) and then performed SA- β -Gal staining. The results indicated that specific inhibition of Erk1/2 had no significant effect on SA- β -Gal activity in SPC, as detailed below:

On the other hand, we added more labels to the ranking scatter plot of siRNA sub-library screening related to p53 activation, as detailed below:

The results of the siRNA library screening experiment suggest that the attenuation of Erk1/2 (*Mapk3/Mapk1*) does not have a significant effect on the activation of p53 within cells, as well as *Tgfb1*.

The above experimental results can preliminarily indicate that the role of p38/p53/Tgf β in cellular senescence and differentiation does not depend on Erk.

We have refined and revised the corresponding sections of the manuscript and highlighted them in red. (See Figure 4a, Figure 5i, and Results (page 6 lines 189 to 193, 211 to 216))

Thank you again for your constructive comments.

2. There are few other points that could be improved and clarified.

1) Regarding the analysis of senescence-related proteins (L118-128, Fig 1g-1, Fig S2) it is not clear which cells were isolated and how was verified the absence of bone cells. The cited reference #49 makes clear that is an easy to mishandle process. Some expression data should be provided to indicate the absence or at least the comparable abundance of non-mesenchymal stem cells in the samples.

The author's answer:

Thank you very much for your comments. We sincerely apologize for not clearly describing the details and procedures of RNA-seq and Western blotting sampling in Fig 1 and Fig S2 in the manuscript. As the reviewer mentioned, we referred to the study by *Maruyama et al.*, which indicated that their procedure for obtaining cranial suture stem cells might be influenced by cranial bone cells and bone marrow-derived stem cells. However, in this study, during the initial design and experimental exploration, we did not plan to exclude the influence of cranial bone cells adjacent to the sutures, which is difficult to avoid (during cranial suture tissue sampling), because we could only sample as close as possible to the coronal suture, using the coronal suture as the midline, with a final sampling width of approximately 2–2.5 mm, i.e., about 1 mm on each side.

In other words, the tissue cells used in RNA-seq may include cranial suture mesenchymal cells as well as some cranial bone cells. After investigating the differential expression of genes in the coronal suture tissue cells of wild-type and mutant mice, we preliminarily found through enrichment analysis that there may be cellular senescence in the coronal sutures of mutant mice. Therefore, we further conducted semi-quantitative protein analysis on the coronal suture tissues of the two groups of mice. Subsequently, after quantitative analysis of senescence-related proteins also showed differences, we continued with immunostaining to clarify the spatial distribution of potential senescent cells.

Combined with follow-up experimental results, this suggests that the main components of cellular senescence may be progenitor cells within the cranial suture mesenchyme.

2) The quantitative results in several plots are given as “area fraction” or “absolute number” (in a suture). Given the differences in the suture size the absolute number or the area fraction do not necessarily reflect the quantitative differences in the state of the suture cells and can easily be over or under evaluated. Percentage of suture mesenchymal cells should be used throughout.

The author's answer:

Thank you for the nice suggestion. We believe it is a very important suggestion. We have re-analyzed and revised the corresponding content.

3) In figure 3b it would appear the CD200+ population in the mutant animals remains clearly distinct from the CD200- suggesting the maintenance of their original characteristics from earlier time points, albeit the overall decrease in cell numbers which would be hard to consolidate with the apparent change in cell cycle in figure 3d. Is the cell cycle difference observed in all time points?

The author's answer:

Thank you for your comment. As the reviewer mentioned, the CD200-positive cell population in the mutant individuals still shows substantial differences compared to the CD200-negative population. We attempted to examine the cell cycle distribution of SPC at other time points. Due to early ossification of the cranial sutures and severe depletion of suture mesenchyme in the mutant individuals, it is difficult to obtain sufficient SPC from mutant mice at postnatal day 10 or later for cell cycle analysis. We additionally analyzed the cell cycle distribution of SPC from mice at postnatal day 7 (P7), and the results also showed significant differences in the G1/G0 and G2/M phase distribution.

We added and improved in the corresponding Results and figures (see figure 3d).

4) The large osteo nodules in figure S4b in the mutant maybe indicative or an increased earlier proliferation state followed by the differentiation state. Is the kinetics of ALP and alizarin read staining in WT ant mutant similar to the endpoint shown?

The author's answer:

Thank you for your meaningful comments. The ALP and alizarin red staining dynamics of the wild-type and mutant are similar to the endpoints shown, that is, osteogenesis in the coronal suture of MUT individuals is enhanced (at least before fusion, as the previous studies showed). Additionally, it should be noted that SPC derived from either WT or MUT may exhibit large calcium nodules after osteogenic induction, but not all samples do. As the reviewer mentioned, the presence of large calcium nodules may indicate an enhanced proliferative state during the differentiation process of the cells, or an increased capacity of osteoblasts to secrete extracellular matrix (or dependent on increased osteoblast proliferation?). We apologize that we cannot yet determine the details or underlying mechanisms of osteoblast lineage differentiation. We tend to speculate that SASP plays an important role in this, thereby promoting osteoblast secretion or enhancing their proliferative expansion.

5) Is hard to tell with the 3-color images of figure 3m the p21 positive cells appear to be much more

than 8-12% shown in figure 2c. Is the variation so large? Maybe single (B&W) images for each channel would be more informative.

The author's answer:

Thank you for raising this question. We noticed this situation during the analysis of experimental images and data. We speculate that the main reason is the change of the primary antibody. When performing Prrx1 and p21 co-staining, we tried using a mouse-derived anti-Prrx1 antibody but did not achieve the desired staining effect. Therefore, we chose to use a more widely used and validated rabbit-derived antibody. Considering the species combination of the two antibodies, during co-staining, we used a mouse-derived primary anti-p21 antibody instead of the rabbit-derived primary anti-p21 antibody used in Fig. 2 (See the supplementary table for specific antibody manufacturers and catalog numbers). For the internal consistency and interpretability of the experimental data, we used the same intensity and voltage of excitation fluorescence during observation and recording. However, due to differences in antibody epitopes, specifications, and production methods, there might be differences in staining performance. We sincerely apologize for this, but we believe it did not affect the interpretability of the results, nor did it affect the stability of the co-localization results.

6) The findings indicate that Tgf-beta may be responsible for the phenotype. Is Tgf-beta1 treatment alone, sufficient to induce synostosis in calvaria explant cultures?

The above points should better elucidate the processes affected and the relevant time point.

The author's answer:

Thank you for your interesting and helpful suggestions. We supplemented this with experiments, treating wild-type mouse calvarial explants with recombinant Tgf- β at different gradient concentrations (0, 0.1, 1 and 10 ng/mL). The results indicated that Tgf- β alone could not induce coronal

suture fusion in vitro. The histological image of the explant coronal suture is shown as follows:

We supplemented the result in the Results sections, see figure 7a, and page 9 lines 312 to 315.

3. Finally, given the focus of the manuscript in the role of senescence in craniosynostosis, a small paragraph or few sentences could be added in the introduction or the discussion addressing the role of senescence in bone mesenchymal differentiation and how this relates to p38 but also Ras/Erk overactivation.

The author's answer:

Thank you for the excellent suggestion. We fully agree that placing our findings in the broader context of signaling pathways, particularly discussing the role of senescence in bone mesenchymal differentiation and the potential crosstalk between p38 MAPK and Ras/Erk pathways, would significantly enrich the discussion. Accordingly, we have added a new paragraph in the Discussion section (please see the last paragraph, which was highlighted in red) to address this point. This paragraph connects our findings on p38 MAPK-driven senescence with the well-established Ras/Erk overactivation in craniosynostosis, thereby providing a more comprehensive explanation for the phenotypes.

We sincerely appreciate the time and effort that you have invested in evaluating our manuscript. We are eager to receive any additional feedback or suggestions that may further enhance our work.

Reviewer #2 (Remarks to the Author):

We feel great thanks for your professional review work on our article. As you are concerned, there are several problems that need to be addressed. According to your nice suggestions, we have made extensive corrections to our previous draft, the detailed corrections are listed below.

Major Comments

1. This work convincingly demonstrates a role for senescence in craniosynostosis. However, the discussion section needs to be enriched to increase this work's impact further. I recommend citing and discussing the following topics: MSC senescence and combating strategies, spatial distribution of Gli1+ SPCs in craniofacial region), (SPCs and craniofacial development, inflammaging of SPC niche, a concise overview of SPCs.

The author's answer:

We sincerely thank you for your positive assessment of our work and for these highly constructive suggestions. The recommended topics are profoundly insightful and serve to effectively contextualize our findings within the broader fields of stem cell biology and skeletal aging, thereby significantly enhancing the impact of our study. We have thoroughly revised the Discussion section accordingly, addressing each of the points you raised (please see the second to last paragraph of the revised manuscript, which was highlighted in red).

2. Although inhibition of p38 α MAPK is effective, the discussion should more carefully address the risks of impairing physiological osteogenesis, as seen in the Mapk14 cKO mice with delayed mineralization (or please present more related data if possible). Long-term effects and safety

considerations for translational therapies should be emphasized.

The author's answer:

We sincerely thank the reviewer for raising this very important and insightful point. We fully agree that the clinical translation of any therapeutic strategy must carefully balance efficacy with safety, particularly regarding potential impacts on physiological processes such as normal bone mineralization. Following the reviewer's suggestion, we have now added a dedicated paragraph in the Discussion section to thoroughly address the potential risks of p38 α MAPK inhibition on physiological osteogenesis, the underlying mechanisms, and the implications for future therapeutic development. We emphasize the need for future studies to explore more selective or transient intervention windows, combination therapies, and long-term safety assessments. We believe this addition significantly strengthens the scientific rigor of our work (please see Discussion section page 13 lines 477 to 494, which was highlighted in red).

3. This work highlights TGF- β 1 as a key SASP factor mediating osteogenic differentiation. However, other SASPs, for example, IL-6, VEGFA, and CXCL12, were upregulated at least at the transcription level. Expanding the discussion on whether these could synergize with TGF- β 1 would strengthen the conclusions. Moreover, TGF- β 1 has been found to be implicated in other types of somatic stem cell senescence, for example, TGF- β 1-induced periodontium stem cell senescence, TGF- β 1-induced AT2 cell and stem cell in lung senescence. Please expand the discussion of the effect of TGF- β 1 in a broader scope and/or context.

The author's answer:

We sincerely thank the reviewer for these insightful and constructive suggestions. We fully agree that contextualizing our findings within the broader network of SASP factors and the role of TGF- β 1 in

various somatic stem cell senescence models will significantly enhance the impact of our study.

Accordingly, we have added a new paragraph to the Discussion section (page 11, lines 433 to 453):

We have systematically discussed how other upregulated SASP factors, such as VEGFA, and CXCL12, may synergize with TGF- β 1 to drive the pathogenesis of craniosynostosis.

We have expanded the discussion to include the established roles of TGF- β 1 in the senescence of somatic stem cells in diverse tissues, such as the lung, periodontium, and heart, thereby positioning our findings within a broader biological context.

We believe these revisions substantially strengthen the depth and scope of the discussion and further highlight the general significance of our discoveries.

Minor Comments

1. Please double-check all abbreviation usage for consistency.
2. Some sentences in the manuscript are too long; please consider breaking them into shorter statements to improve clarity and delivery.
3. Figures S6 and S7 contain critical mechanistic evidence; please consider moving parts of them into the main figures.

The author's answer:

We sincerely thank the reviewer for the thorough reading of our manuscript and these very helpful comments. We have carefully revised the manuscript according to each of the suggestions, as detailed below:

- (1) Regarding abbreviation consistency: We have systematically checked all abbreviations throughout the manuscript to ensure that each is defined upon its first appearance and used consistently thereafter.
- (2) Regarding long sentences: We have thoroughly reviewed the text and have broken down overly

long sentences into shorter ones to significantly improve clarity and readability.

(3) Regarding figure reorganization: We fully agree that the data in Figures S6 and S7 provide critical mechanistic evidence. Accordingly, we have moved key data from the original Figure S6 and Figure S7 into the main Figure 8 and Figure 9, respectively, to make these central arguments more prominent. We believe these revisions have substantially improved the quality of the manuscript. Thank you again for your valuable time and guidance.

Recommendation

This is a high-quality and innovative work that significantly advances the mechanistic understanding of craniosynostosis. With improved contextualization in the relevant literature, expanded discussion of safety and translation, and minor editorial refinements, it will be well-suited for publication in *Communications Biology*.

The author's answer:

Thank you for the nice and helpful suggestions. We sincerely appreciate the time and effort that you have invested in evaluating our manuscript. We are eager to receive any additional feedback or suggestions that may further enhance our work.

Reviewer #3 (Remarks to the Author):

The manuscript addresses a relevant and timely aspect of craniosynostosis pathophysiology by identifying the role of the senescence process as a central mechanism driving the abnormal behavior of mesenchymal stromal cells within the cranial suture niche. It also explores a potential therapeutic target, thereby offering alternative strategies for treatment. The work is clearly written and of interest to the field. Only minor revisions are required to further strengthen the study.

The author's answer:

We feel great thanks for your professional review work on our article. As you are concerned, there are several problems that need to be addressed. According to your nice suggestions, we have made extensive corrections to our previous draft, the detailed corrections are listed below.

Specific comments:

1. The Discussion could be strengthened by including an overview of therapeutic strategies currently explored in vitro, with particular attention to those targeting FGFR2 variants (e.g., DOI: 10.1016/j.omtn.2024.102427).

The author's answer:

We sincerely thank you for this valuable suggestion. We agree that including an overview of therapeutic strategies targeting FGFR2 variants in the Discussion section better highlights the potential translational value of our research. Following your guidance, we have added a new paragraph to the Discussion (pages 13 to 14, lines 502 to 527) summarizing current exploratory therapeutic strategies targeting FGFR2 variants in in vitro studies, including small-molecule inhibitors and monoclonal antibodies, with specific citation of the recommended literature (DOI: 10.1016/j.omtn.2024.102427).

We believe this addition significantly strengthens the depth and scope of the Discussion.

2. The authors are encouraged to elaborate in the Introduction or Discussion on the rationale for selecting shRNA over a pharmacological approach, especially in view of the therapeutic window supported by transient RNAi-based molecules.

The author's answer:

We sincerely thank you for this insightful comment. We agree that clarifying the rationale for selecting a genetic approach (shRNA) over pharmacological inhibitors is crucial to highlight the potential of our therapeutic strategy. Following the reviewer's suggestion, we have added corresponding content in the Discussion section (page 13, lines 466 to 476). We believe this addition significantly enhances the logical coherence and persuasiveness of our research strategy.

3. The Results section does not consider recent studies showing that DDR2+ mesenchymal stem cell subpopulations within cranial sutures represent primary drivers of pathological premature ossification in craniosynostosis (DOI: 10.1038/s41586-023-06526-2). This aspect should be addressed and its implications for the findings discussed.

The author's answer:

We sincerely thank you for this valuable suggestion and for bringing to our attention the important 2023 study by Greenblatt's team published in Nature. We fully agree that connecting our findings with the newly identified pathological driver - the DDR2+ mesenchymal stem cell subpopulation - is crucial for improving the narrative logic and scientific depth of our study. Accordingly, we have added a new paragraph in the Discussion section (page 11, lines 417 to 429) to thoroughly explore the potential synergistic relationships and mechanistic connections between these two aspects. We believe this

revision has significantly enhanced both the completeness and impact of our manuscript.

4. The authors should evaluate whether specifically targeting the mutant allele of the disease-causing gene in gain-of-function conditions (e.g. Crouzon syndrome) could restore cell cycle progression, reduce the senescence phenotype, and minimize off-target effects.

The author's answer:

We sincerely thank you for this profoundly insightful and forward-looking comment. In the context of gain-of-function mutations (such as those in FGFR2 causing Crouzon syndrome), evaluating strategies for specifically targeting the mutant allele is crucial for developing highly effective and safe therapies. This precision medicine approach minimizes interference with the wild-type allele, thereby theoretically significantly reducing off-target risks. For craniosynostosis syndromes driven by specific FGFR2 point mutations (e.g., C342Y/S252W), this strategy holds exceptional appeal. By designing allele-specific RNAi (such as siRNA or shRNA) targeting the mutant sequence or exploring CRISPR-mediated gene editing technologies, it becomes theoretically feasible to selectively silence or correct the mutant FGFR2 allele while completely preserving the normal function of the wild-type allele. Such precise intervention could fundamentally reverse downstream signaling pathways (e.g., p38 MAPK and ERK) hyperactivated by the mutant protein, thereby directly restoring abnormal cellular phenotypes. More importantly, since wild-type FGFR2 is indispensable for physiological processes including skeletal development and phosphate metabolism, preserving its function is essential for minimizing off-target effects and long-term adverse consequences. This provides a potential new approach to addressing the issue of impaired physiological osteogenesis observed in our Mapk14 cKO model. Although challenges remain in achieving efficient in vivo delivery and ensuring absolute allele specificity, this strategy represents an important future direction for gene therapy in craniosynostosis

(Discussion section, pages 13 to 14, lines 511 to 527).

On the other hand, our results indirectly suggest a relationship between Fgfr2 gain-of-function mutations and the cellular senescence phenotype. Using SPCs derived from wild-type individuals and transducing them with lentiviral vectors carrying Fgfr2 point mutations, we observed the emergence of cellular senescence phenotypes, thereby demonstrating a causal relationship between the two to some extent.

Thank you again for your very helpful comments!

5. The potential application of functionalized nanoparticles for cell-specific targeting should also be evaluated, or at least discussed, as this approach could provide additional translational relevance

The author's answer:

We sincerely thank you for this highly insightful and forward-looking suggestion. We fully agree that evaluating the use of functionalized nanoparticles as a delivery platform for achieving cell-specific targeting would significantly enhance the translational relevance and clinical feasibility of our therapeutic strategy. Accordingly, we have added a dedicated paragraph in the Discussion section (page 13, lines 494 to 505) to evaluate and discuss the considerable potential of utilizing nanoparticles (e.g., lipid nanoparticles or polymeric nanoparticles) to encapsulate therapeutic RNAi (e.g., shRNA) or gene-editing machinery for precision delivery. We believe this addition not only directly addresses the reviewer's point but also charts a promising technological path for translating our proposed therapeutic concept from bench to bedside.

6. Consistency in the use of past and present tense throughout the manuscript should be ensured.

The author's answer:

We sincerely thank you for the careful reading of our manuscript and for raising this important point. Ensuring consistency in verb tense throughout the manuscript is crucial for its clarity and professionalism. We have performed a thorough check and revision of the entire manuscript (including the Abstract, Methods, Results, and Discussion) to standardize tense usage.

7. The representative images of ALP staining (e.g. figures 5g-5h, 6h, 7d-l etc) should be replaced with higher-quality ones.

The author's answer:

We sincerely thank you for pointing this out. High-quality representative images are crucial for clearly presenting the experimental results. As suggested, we have repeated the ALP staining assays and have replaced all the indicated images (Figures 5g-h, 6h, and 7d-l) with new ones of higher resolution and better contrast. These new images, while maintaining the original scientific conclusions, significantly improve the visual quality and clarity. The revised manuscript has been updated accordingly. Thank you again for helping us enhance the overall quality of the paper.

8. The Discussion should include a more detailed explanation of p38 signaling, clarifying its role as a downstream effector of FGFR2 or its relationship with Twist pathways.

The author's answer:

We sincerely thank you for this valuable suggestion. We agree that a clearer elucidation of the relationship between the p38 signaling pathway and core pathogenic pathways in craniosynostosis (such as FGFR2 or Twist) in the Discussion (pages 13 to 14, lines 506 to 531) is crucial for deepening the mechanistic insights of our study. Accordingly, we have added in the Discussion section that details the potential role of p38 MAPK as a downstream effector of FGFR. We believe this addition

significantly strengthens the logical flow and mechanistic depth of the Discussion.

We sincerely appreciate the time and effort that you have invested in evaluating our manuscript. We are eager to receive any additional feedback or suggestions that may further enhance our work.